# Phosphorylation of tau at a single residue inhibits binding to the E3 ubiquitin ligase, CHIP

Cory M. Nadel [1,2], Saugat Pokhrel [1,2], Kristin Wucherer[1], Abby Oehler[2], Aye C. Thwin [2,3], Koli Basu[1], Matthew D. Callahan [1,4], Daniel R. Southworth [2,4], Daniel A. Mordes [3,4], Charles S. Craik [1] & Jason E. Gestwicki [1,4] ✉

Microtubule-associated protein tau (MAPT/tau) accumulates in a family of neurodegenerative diseases, including Alzheimer's disease (AD). In disease, tau is aberrantly modified by post-translational modifications (PTMs), including hyper-phosphorylation. However, it is often unclear which of these PTMs contribute to tau's accumulation or what mechanisms might be involved. To explore these questions, we focus on a cleaved proteoform of tau (tauC3), which selectively accumulates in AD and was recently shown to be degraded by its direct binding to the E3 ubiquitin ligase, CHIP. Here, we find that phosphorylation of tauC3 at a single residue, pS416, is sufficient to weaken its interaction with CHIP. A co-crystal structure of CHIP bound to the C-terminus of tauC3 reveals the mechanism of this clash, allowing design of a mutation (CHIP[D134A]) that partially restores binding and turnover of pS416 tauC3. We confirm that, in our models, pS416 is produced by the known AD-associated kinase, MARK2/Par-1b, providing a potential link to disease. In further support of this idea, an antibody against pS416 co-localizes with tauC3 in degenerative neurons within the hippocampus of AD patients. Together, these studies suggest a molecular mechanism for how phosphorylation at a discrete site contributes to accumulation of a tau proteoform.

The deposition of microtubule-associated protein tau (*MAPT*/tau) into insoluble fibrils is a major pathological hallmark of fatal and incurable neurodegenerative diseases—including Alzheimer's disease (AD), frontotemporal dementia (FTD), progressive supranuclear palsy (PSP), corticobasal degeneration (CBD), and others[1–3]. There is interest in understanding the pathways that govern tau proteostasis, in both health and disease[4], because such knowledge may reveal mechanisms for reducing tau accumulation.

Many studies have suggested that CHIP (C-terminus of Hsc70 interacting protein; *STUB1*) is one of the major E3 ubiquitin ligases responsible for degradation of tau[5]. For example, the CHIP[-/-] mouse accumulates tau[6], CHIP overexpression reduces tau aggregation[7], and a recent unbiased screen identified the CHIP pathway as a key regulator of tau pathogenicity[8]. In the canonical mechanism, CHIP is recruited to tau via the molecular chaperone, heat shock protein 70 (Hsp70)[9]. In the first step, Hsp70 binds to at least two major sites in

[1]Department of Pharmaceutical Chemistry, University of California San Francisco, San Francisco, CA 94158, USA. [2]Institute for Neurodegenerative Diseases, University of California San Francisco, San Francisco, CA 94158, USA. [3]Department of Biochemistry & Biophysics, University of California San Francisco, San Francisco, CA 94158, USA. [4]Department of Pathology, University of California San Francisco, San Francisco, CA 94158, USA. ✉e-mail: Jason.Gestwicki@ucsf.edu

tau's microtubule-binding repeats (MTBRs)[10–14]. Then, Hsp70 uses a C-terminal sequence, termed the EEVD motif, to bind CHIP's tetratricopeptide repeat (TPR) domain[15–19]. In this way, certain isoforms of Hsp70[20] act as adapters, recruiting CHIP and promoting tau's ubiquitination and turnover[12,21]. However, the CHIP-Hsp70 complex does not seem to act on all tau proteoforms in the same way. Tau is found in a large number of proteoforms, including those resulting from alternative splicing and post-translational modifications (PTMs), such as phosphorylation[22]. These different tau proteoforms seem to have distinct ways of interacting with CHIP, which likely impacts their relative rate(s) of turnover. For example, the Hsp70-CHIP complex preferentially recognizes pathologically phosphorylated tau[23,24]. Moreover, some tau proteoforms even bind CHIP without the need for chaperone. For example, while CHIP has relatively poor affinity for unmodified tau[25], it directly interacts with tau that has been phosphorylated at sites in the N-terminal domain (NTD) and proline-rich region (PRR) by glycogen synthase kinase 3β (GSK-3β)[26] Together, these studies show that CHIP interacts with different tau proteoforms using chaperone-dependent and chaperone-independent mechanisms and that these contacts are broadly sensitive to phosphorylation.

Despite these substantial advances, the molecular mechanisms that govern CHIP's recognition of tau proteoforms are not clear. We saw an opportunity to address this question by studying tauC3, a tau proteoform that is the epitope for the widely used C3 antibody. TauC3 is the product of caspase cleavage at aspartate 421 (D421) in tau and multiple lines of evidence suggest that this truncation plays an important role in neurodegenerative disease[27]. For example, immunoreactivity with the C3 antibody strongly correlates with progression of AD[28] and other tauopathies[29]. In addition, the cleavage event to produce tauC3 seems to precede tau deposition[30,31], treatment with the C3 antibody partially blocks tau seeding[32], the tauC3 protein is more aggregation-prone than full-length tau and expression of this proteoform inhibits microtubule dynamics and slows axonal transport in neurons[33–35]. Like other tau proteoforms, tauC3 is known to be a substrate of CHIP. For example, the CHIP[-/-] mouse preferentially accumulates tauC3[6] and this isoform is very rapidly ubiquitinated by CHIP in vitro[36]. We reasoned that the tauC3 system might be a particularly good model for studying the effects of phosphorylation on CHIP binding because the interaction site is comparatively well defined. Specifically, tauC3 has an EEVD-like motif at its C-terminus, which, similar to Hsp70, binds with high affinity to CHIP's TPR domain[36].

Here, we explore how tauC3 phosphorylation impacts its binding to CHIP. This question is important because, despite being an excellent substrate for CHIP, tauC3 accumulates in the brains of AD patients[28], suggesting that unknown factors allow it to evade CHIP-mediated quality control. To explain this apparent dichotomy, we crafted a hypothesis based on data from our group and others, in which phosphorylation of a specific residue in Hsp70's EEVD motif was found to block its binding to CHIP[37,38]. We noticed that tauC3's EEVD-like motif has a serine at the equivalent position, S416 (numbered according to the 2N4R-tau splice isoform); thus, we hypothesized that its phosphorylation might restrict CHIP binding. Here, we apply a multidisciplinary approach to show that, indeed, pS416 is sufficient to weaken the CHIP-tauC3 interaction in vitro and in cells. To understand the molecular mechanism, we used crystallography to show that pS416 creates a steric and electronic clash in CHIP's TPR domain and we leveraged this information to create a CHIP point mutant (D134A) that partially regains the ability to bind and ubiquitinate tauC3 pS416. This mechanism might be important in disease, because we find that pS416 co-accumulates with tauC3 in dysmorphic neurons within the hippocampus of AD patient brains and that the AD-associated[39–41] kinase, microtubule affinity regulating kinase 2 (MARK2/Par-1b), generates pS416. Together, these studies provide insight into how a hierarchical series of tau PTMs—first proteolysis to create tauC3, then phosphorylation by MARK2 to limit CHIP binding and finally CHIP-mediated transfer of ubiquitin—balances tau proteostasis.

## Results

### Phosphorylation of tauC3 inhibits interaction with CHIP

Previous work has shown that phosphorylation of Hsp70's C-terminus weakens binding to CHIP[37,38]. Thus, we hypothesized that a similar mechanism might govern the binding of tauC3's C-terminal degron to CHIP. To test this idea, we first used a live cell NanoBiT split-luciferase assay[42] to measure CHIP-tauC3 interactions in HEK293 cells (Fig. 1A). As a control, we first confirmed[36] that tauC3 interacts better with CHIP than full length (FL) tau under basal conditions (Fig. 1B). Throughout the manuscript, we use FL to refer to the 0N4R splice isoform of tau. Next, we treated with the serine/threonine protein phosphatase inhibitor, okadaic acid (30 nM), and found that it enhanced CHIP's interaction with FL tau, consistent with the literature[23,24], while it significantly lessened the interaction with tauC3 (Fig. 1B). Thus, it seems that phosphorylation has dramatically different effects on FL tau and tauC3, highlighting the proteoform selectivity in CHIP binding. To support these cell-based observations with biochemical studies, we then purified recombinant, natively phosphorylated tauC3 (p.tauC3) protein from Sf9 insect cells (Supplementary Fig. S1A). Consistent with previous studies[23], p.tauC3 produced in this way is heavily phosphorylated, as measured by immunoblots for specific tau phosphoepitopes, including pS202/pT205 (AT8), pS396 (PHF13), and pS416 (Supplementary Fig. S1B) as well as mass spectrometry (Supplementary Fig. S1C). Using this protein and unmodified, recombinant tauC3 collected from E. coli, we performed ELISAs to measure CHIP binding (Fig. 1C, D). The results showed that p.tauC3 binds ~4-fold weaker to CHIP ($K_d = 0.70 \pm 0.12\,\mu M$) than unmodified tauC3 ($K_d = 0.19 \pm 0.02\,\mu M$). To probe how much of this loss of affinity for p.tauC3 might be due to phosphorylation near the C-terminus instead of secondary sites, we also measured binding of CHIP to full length p.tau, produced in the same Sf9 cells. CHIP is known to bind internal phospho-sites[23] and we noted that its affinity for full length p.tau is approximately the same ($K_d$ ~0.5 μM; see Fig. 1C, D) as it's affinity for p.tauC3 ($K_d$ ~0.7 μM), consistent with the idea that secondary phospho-sites also contribute to p.tauC3 binding. We then compared these different tauC3 proteins as substrates for CHIP in ubiquitination reactions in vitro. In these experiments, CHIP rapidly converted tauC3 to high-molecular weight (HMW) polyubiquitinated species; however, this process was modestly attenuated for p.tauC3 (Fig. 1E and Supplementary Fig. S1D). We subsequently identified the ubiquitination sites present on the CHIP-treated tauC3 proteoforms by mass spectrometry (Supplementary Fig. S1E), confirming that both p.tauC3 and tauC3 could be modified by CHIP. To confirm that phosphorylation— and not a different PTM—was responsible for the observed weakening of the CHIP-tauC3 interaction, we treated tauC3 or p.tauC3 with lambda phosphatase, and then compared the binding of these proteins to CHIP using ELISAs. De-phosphorylation of p.tauC3 was confirmed via immunoblot (Supplementary Fig. S1F). Under these conditions, we found that removing the phosphates was sufficient to rescue binding of p.tauC3 to CHIP (Supplementary Fig. S1G). Together, these results indicate that phosphorylation of tauC3 weakens binding to CHIP and hinders the ability of CHIP to ubiquitinate this substrate.

### Phosphorylation of tauC3 Ser416 inhibits CHIP binding

We next sought to identify the phosphorylation site(s) in p.tauC3 that might weaken binding to CHIP. The 20 phosphorylation sites that we had identified on the insect-cell derived p.tauC3 (see Supplementary Fig. S1B) were rather broadly located in the domains of tau. Specifically, 1/20 of the sites are located in the N-terminal domain (NTD), 12/20 in the proline-rich domain (PRD), 3/20 in the microtubule-binding repeats (MTBRs), and 4/20 in the C-terminal domain (CTD). As a first

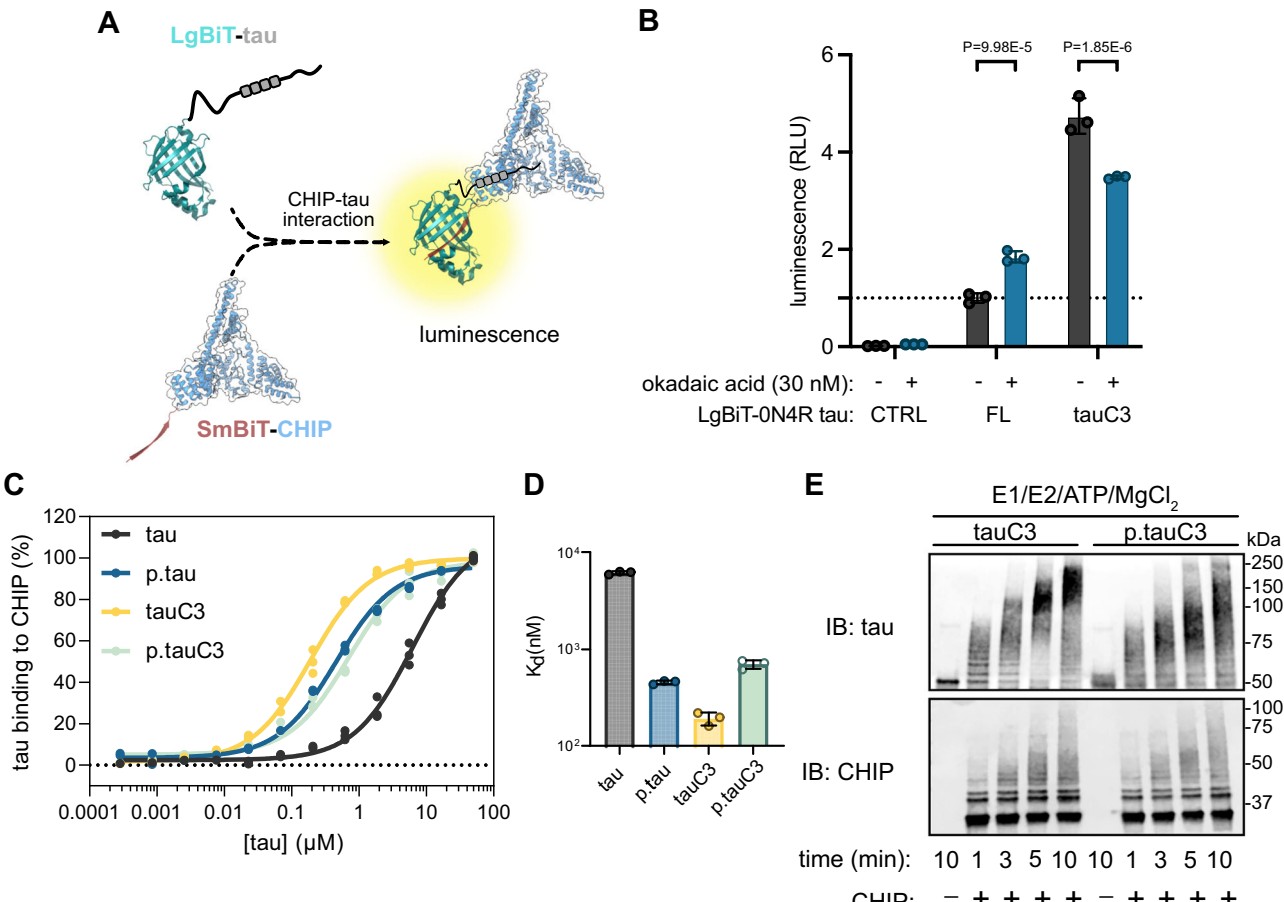

**Fig. 1 | Phosphorylation of tauC3 inhibits interaction with CHIP. A** Cartoon schematic of the live-cell NanoBiT assay for measuring CHIP-tau PPIs (NanoBiT PDB = 7SNX; CHIP PDB = 2C2L). **B** Results of NanoBiT assays for CHIP interactions with tau proteoforms following treatment with okadaic acid (30 nM, 18 h) or DMSO control. Luminescence normalized to CHIP binding to FL 0N4R tau treated with vehicle (dashed line). Error bars represent standard deviation (SD). Statistical significance was determined by two-way ANOVA with Bonferroni's post hoc analysis ($n = 3$ biological replicates). CTRL = mock transfected control. **C** Tau proteoforms binding to immobilized CHIP measured by ELISA. Assay was performed in technical triplicate and normalized to maximum absorbance at 450 nm. **D** Dissociation constants derived from (**C**). Error bars represent SD. **E** In vitro ubiquitination of tau proteoforms by CHIP. Samples were collected at the denoted timepoints, quenched in SDS-PAGE loading buffer, and analyzed by western blot. Assay was performed once.

step at isolating which of these sites might be important, we expressed truncated tau constructs (Fig. 2A) in HEK293 cells and looked for changes in the CHIP interaction after okadaic acid treatment, as measured by NanoBiT assays. Okadaic acid treatment caused nearly all the constructs to bind tighter to CHIP (Fig. 2B), consistent with reports of CHIP binding to phosphorylated tau at multiple sites[26]. However, the one exception was the tauC3 construct, where, consistent with our earlier data, okadaic acid treatment weakened binding. This result re-enforced the idea that tau proteoforms interact with CHIP through distinct mechanisms and suggested that the key, inhibitory phosphorylation site(s) for tauC3 are located near the C-terminus. Our mass spectrometry data showed only four phosphosites present in this region—S396, S400, S404, and S416 (see Supplementary Fig. S1E). Due to previous work on analogous Hsp70 phosphosites[37], we hypothesized that tau's S416 was likely to play an important role. Specifically, phosphorylation of residue T636 in the C-terminus of Hsp70 (HSPA1A) is known to weaken binding to CHIP's TPR domain, by creating a steric and electronic clash (Supplementary Fig. S2A, B)[37]. To confirm this finding, we used 10mer peptides corresponding to the C-terminus of Hsp70 and measured binding to CHIP using differential scanning fluorimetry (DSF) and fluorescence polarization (FP). Consistent with previous experiments, phosphorylation of pT636 or replacing that residue with the pseudo-phosphorylation mimetic, T636E, weakened binding by either DSF (Figs. 2C and S2C) or FP (Fig. 2E, F). When we

aligned the C-termini of Hsp70 and tauC3, we noted that T636 of Hsp70 corresponds to S416 in tauC3 (Fig. S2D), leading us to hypothesize that this residue in tauC3 might behave similarly. To test this idea, we synthesized 10-mer acetylated peptides corresponding to the C-terminus of tau (Ac-SSTGSIDMVD-OH) and then replaced each of the possible Ser/Thr residues, including S416, with glutamic acid. When these peptides were tested for binding to CHIP, we found that only the pS416 phosphomimetic (Ac-SSTGEIDMVD-OH), and not any of the others, was sufficient to weaken binding by DSF (Supplementary Fig. S3A, B) or FP (Supplementary Fig. S3C). Specifically, by DSF, we observed an -6 °C stabilization of CHIP in the presence of WT tauC3 peptide compared to DMSO control (CHIP $Tm_{app}$ DMSO = 42.15 ± 0.22 °C; SSTGSIDMVD = 48.81 ± 0.11 °C) and this binding was weakened by the introduction of the phosphomimetic mutation (CHIP $Tm_{app}$ SSTGEIDMVD = 45.49 ± 0.11 °C) (Figs. 2D and S3D). In FP assays, the phosphomimetic tauC3 peptide was also a weaker competitor than WT tauC3 ($K_i$ WT tauC3 = 0.15 ± 0.02 μM; tauC3 S416E = 2.41 ± 0.33 μM) (Fig. 2E, F). Moreover, we introduced a bona fide phosphoserine residue at the S416 position (tauC3 pS416) and likewise observed much weaker competition by this peptide using either DSF (see Fig. 2D and Supplementary Fig. S3D) or FP (see Fig. 2E, F; $K_i$ tauC3 pS416 = 1.01 ± 0.99 μM). To test whether this single site was important in the context of the tauC3 protein (and not just the 10mer peptide), we purified tauC3 with a single phosphomimetic mutation (S416E) and tested its

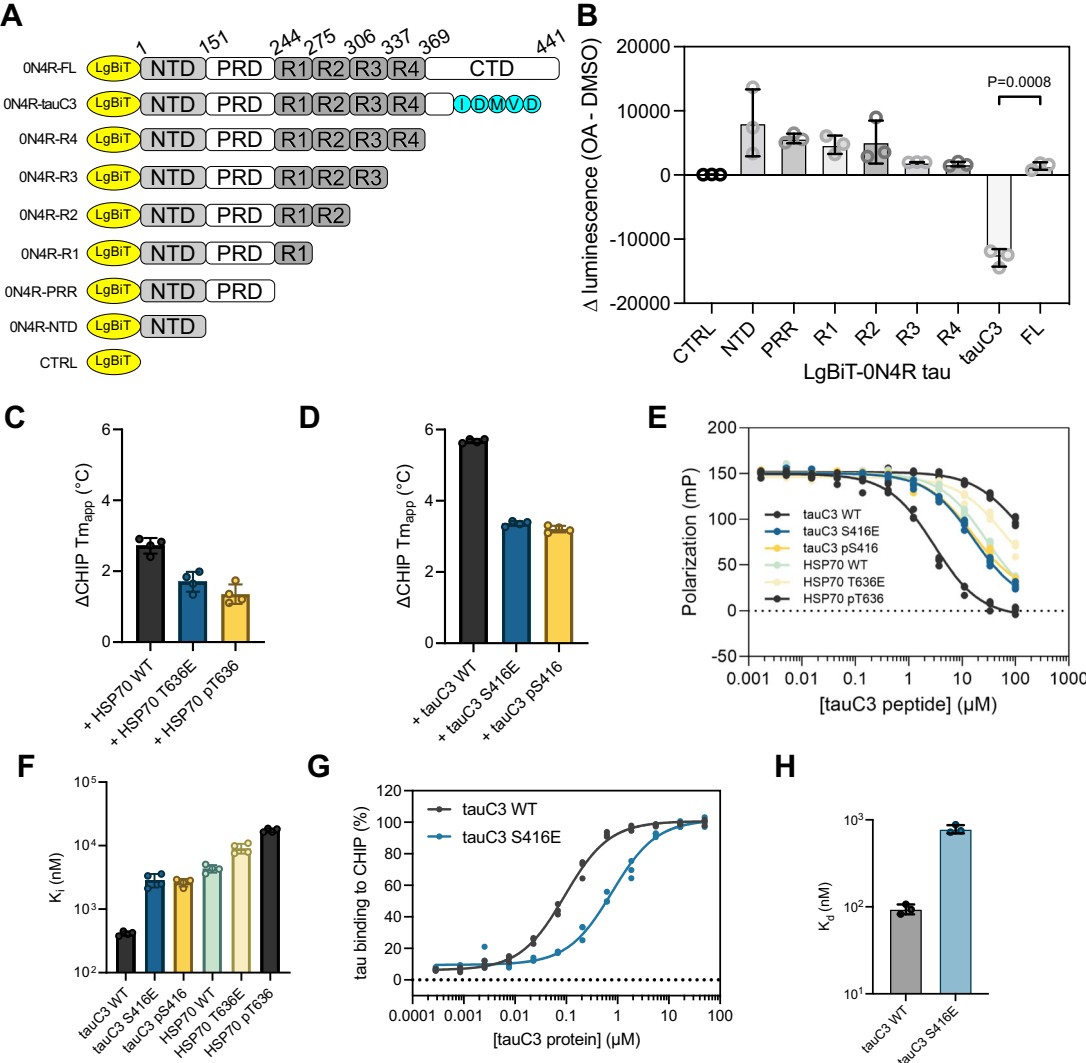

**Fig. 2 | Phosphorylation of tauC3 Ser416 is sufficient to inhibit interaction with CHIP. A** Cartoon depicting constructs used for mapping of inhibitory phosphorylation sites by NanoBiT live cell assays. **B** NanoBiT live cell assay for CHIP interactions with truncated tau constructs following treatment with okadaic acid (30 nM, 18 h) or DMSO control. Data are shown as the difference between okadaic acid treated samples and vehicle control. Error bars represent SD. Statistical significance was determined by two-sided, one-way ANOVA with Tukey's post hoc analysis ($n = 3$ biological replicates). **C** Apparent change in melting temperatures ($Tm_{app}$) for CHIP incubated with DMSO control or Hsp70 C-terminal peptides, as determined by DSF. Error bars represent SD. ($n = 4$ technical replicates). **D** DSF experiments performed and analyzed as in the previous panel, but with tauC3-derived peptides. ($n = 4$ technical replicates). **E** Competition FP experiment showing displacement of fluorescent tracer from the CHIP TPR domain by various 10-mer Hsp70 or tauC3-derived peptides. Experiments were performed in technical quadruplicate. **F** Inhibition constants derived from (**E**). Error bars represent SD. **G** Tau proteoforms binding to immobilized CHIP, as measured by ELISA. Assay was performed in technical triplicate and normalized to maximum absorbance at 450 nm. **H** Dissociation constants derived from (**G**). Error bars represent SD.

binding to CHIP using ELISA (Fig. 2G). Consistent with the results from the peptide experiments, tauC3 S416E bound weaker to CHIP compared to WT tauC3 ($K_d$ WT = 0.09 ± 0.01 µM; S416E = 0.78 ± 0.08 µM) (Fig. 2J). Indeed, the effect of S416E on CHIP binding was nearly the same as the heavily phosphorylated p.tauC3 protein (see Fig. 1), suggesting that this single PTM is necessary and sufficient to weaken the CHIP-tauC3 interaction.

**TauC3 Ser416 phosphorylation regulates its homeostasis**

We hypothesized that S416 phosphorylation might also slow CHIP-mediated ubiquitination of tauC3. To test this idea, we performed ubiquitination reactions in vitro. As previously observed, CHIP robustly ubiquitinated tauC3 to a greater extent than FL tau (Fig. 3A, B). However, this activity was attenuated by the S416E phosphomimetic mutation (Fig. 3A, B). It is worth noting that CHIP retained some activity on tauC3 S416E, consistent with the weakened, but not completely blocked, affinity (see Fig. 2J). Consistent with the idea that

affinity corresponds to relative ubiquitination rates, previous studies have shown that mutating the terminal aspartate in tauC3 abolishes binding to CHIP and largely prevents in vitro ubiquitination[36]. Next, we hypothesized that reduced ubiquitination might partially protect tauC3 from degradation in cells. To test this idea, we generated HEK293 Flp-In lines that express green fluorescent protein (GFP)-tagged tau variants from a doxycycline-inducible promoter (Fig. 3C). Importantly, this platform involves integration of the transgene at a single shared genomic locus, reducing expression variability introduced by transfection. Using microscopy, we first confirmed that FL tau, tauC3 and tauC3 S416E all localized properly to the microtubules (Fig. 3D). Next, we tested their binding to endogenous CHIP by co-immunoprecipitation. Consistent with the in vitro studies, we found that tauC3, but not FL tau, strongly co-immunoprecipitated endogenous CHIP from cells (Fig. 3E). Moreover, this binding was partially inhibited by the S416E phosphomimetic mutation (relative amount = 0.55). In these cells, we noticed that the overall levels of tauC3 protein

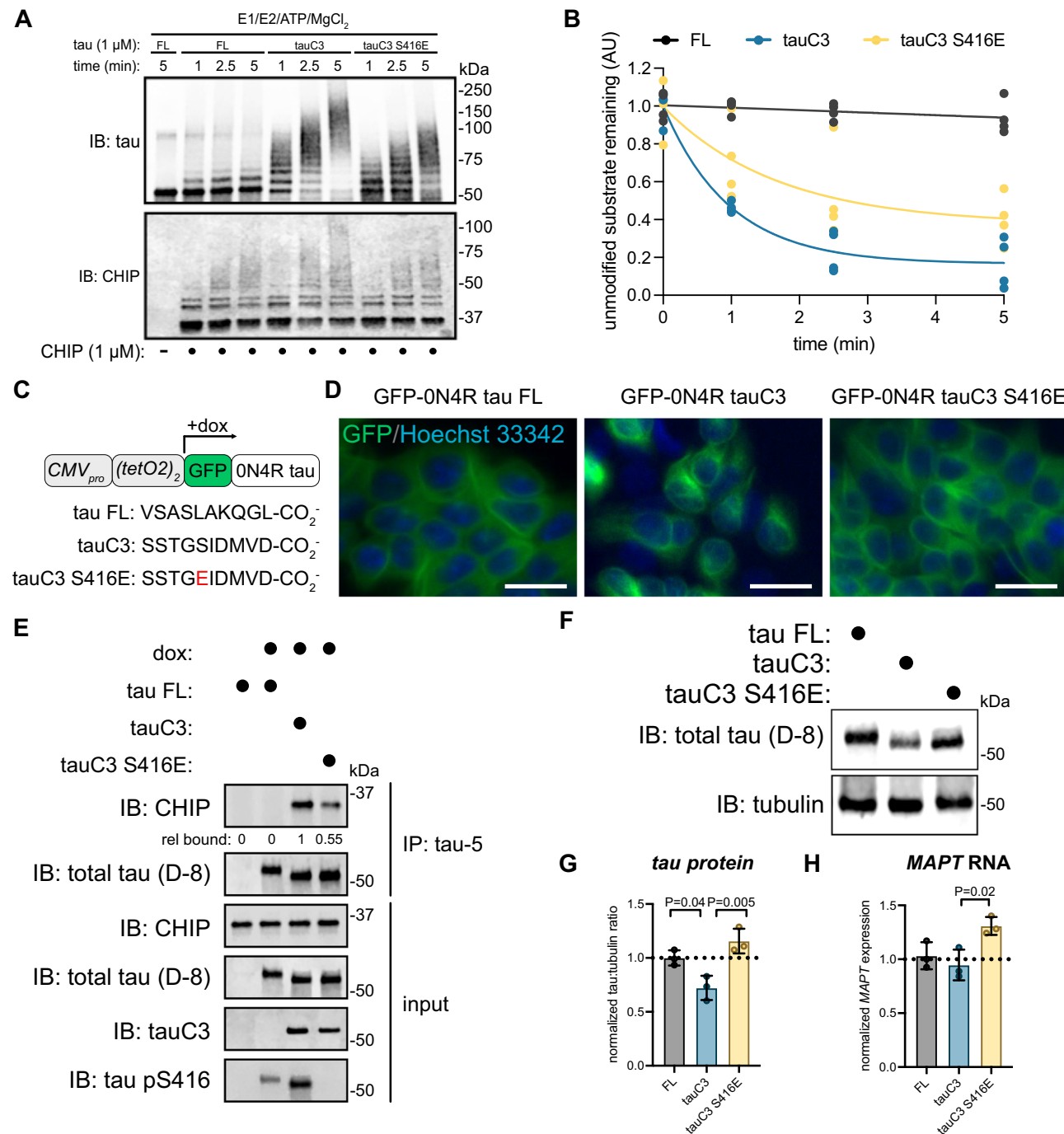

**Fig. 3 | TauC3 Ser416 phosphorylation regulates CHIP-dependent tau home-ostasis. A** In vitro ubiquitination of tau proteoforms by CHIP. Samples were collected at the denoted timepoints, quenched in SDS-PAGE loading buffer, and analyzed by western blot. **B** Quantification of unmodified tau remaining following in vitro ubiquitination of varying tau proteoforms by CHIP. Unmodified tau remaining was analyzed by densitometry, normalized to 0-min time point, and curves were fit using one-phase exponential decay ($n = 3$ technical replicates). **C** Cartoon schematic depicting promoter architecture and varying C-terminal sequences for HEK293 FlpIn T-Rex cells expressing doxycycline inducible GFP-tau proteoforms. **D** Representative fluorescence micrographs for GFP-tau cells. Images show tau species associated with microtubules (green, false color), while nuclei are stained with Hoechst 33342 (blue, false color). Scale bar = 30 μm. Images are representative of three separate acquisitions. **E** Co-

immunoprecipitation assay following IP of varying tau species from cells. Whole cell lysate (input) was used for loading controls, and co-immunoprecipitated CHIP was analyzed by western blot. Experiment was repeated in duplicate. **F** Representative western blot showing differing abundance of various tau pro-teoforms in HEK293 FlpIn T-Rex cells. Experiment was repeated in duplicate. **G** Quantification of tau protein abundance taken from three independent biolo-gical experiments. Tau:tubulin ratio was determined by densitometry and nor-malized to full-length tau. Error bars represent SD. Statistical significance was determined by one-way ANOVA with Tukey's post hoc analysis. **H** Quantification of *MAPT* mRNA from three independent biological experiments. *MAPT* mRNA was normalized to *GAPDH* and shown relative to full-length tau. Error bars represent SD. Statistical significance was determined by one-way ANOVA with Tukey's post hoc analysis.

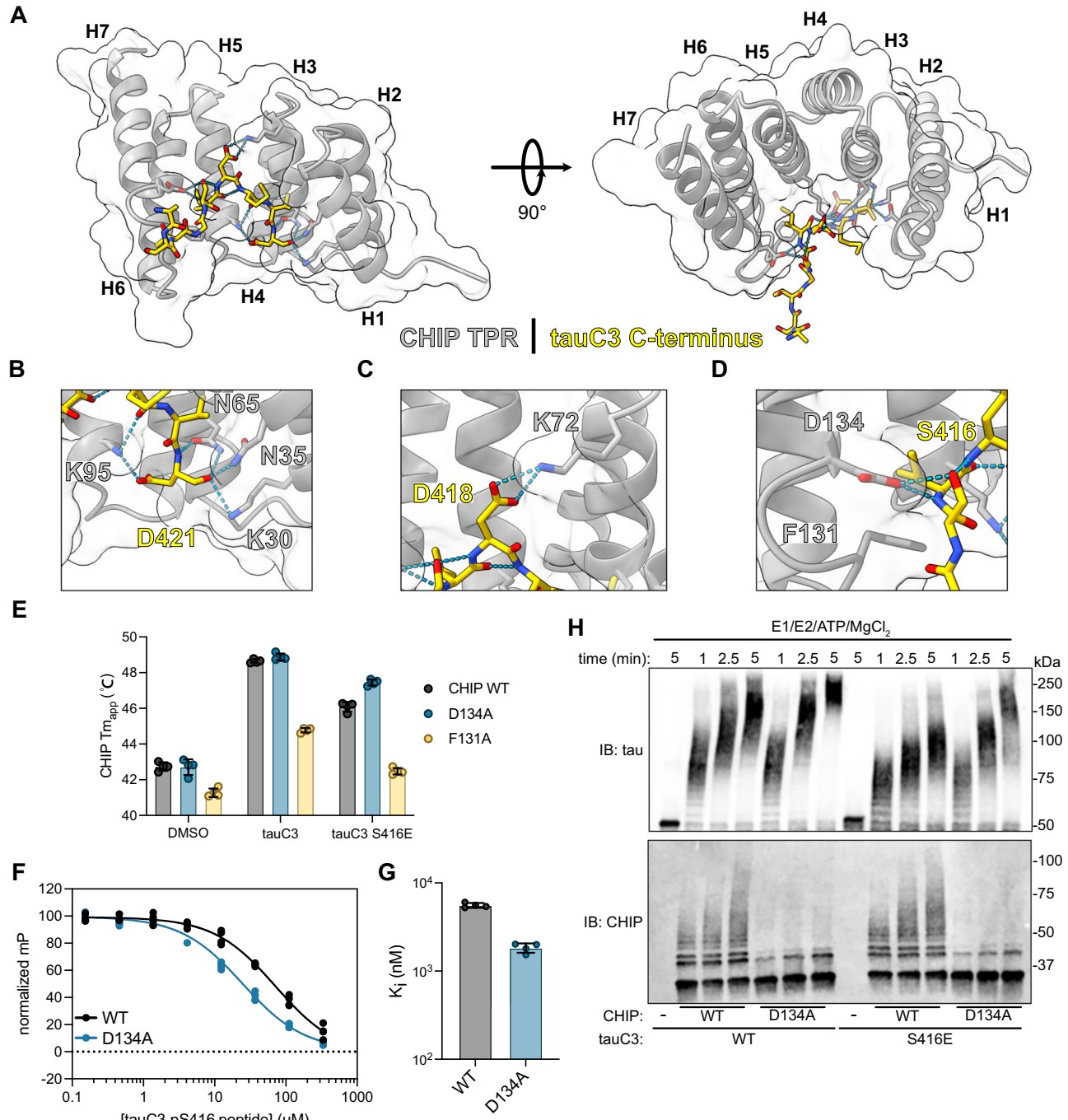

**Fig. 4 | Structural basis for CHIP binding to tauC3 and inhibition by phosphorylation. A** 1.8 Å crystal structure of the CHIP TPR domain bound to a 10-mer tauC3 C-terminal peptide. The CHIP TPR domain is depicted in gray, while the tauC3 peptide is in yellow. Helices 1-7 of the CHIP TPR domain are numbered H1-H7. **B** Close-up view of interactions of tauC3 D421 and C-terminus with CHIP carboxylate-clamp residues K95, N65, N35, and K30. **C** Close-up view of interaction of tauC3 D418 with CHIP carboxylate-clamp residue K72. **D** Close-up view of interactions of tauC3 S416 with CHIP carboxylate-clamp residue D134 and proximity to CHIP F131. **E** Apparent melting temperatures ($Tm_{app}$) of CHIP WT or mutants in the absence or presence of 10-mer tau peptides as derived from DSF experiments. Error bars represent SD. ($n$ = 4 technical replicates). **F** Competition FP experiment showing displacement of fluorescent tracer from WT or D134A CHIP TPR domain by tauC3 pS416 10-mer peptide. Samples were performed in technical quadruplicate and normalized to DMSO control. **G** Inhibition constants for tauC3 pS416 peptide for WT or D134A CHIP as derived from (**F**). Error bars represent SD. **H** In vitro ubiquitination of tau proteoforms by WT or D134A CHIP. Samples were collected at the denoted timepoints, quenched in SDS-PAGE loading buffer, and analyzed by western blot.

were modestly lower relative to that of FL tau or tauC3 S416E (Fig. 3F, G). The reduced levels of tauC3 were not due to transcriptional effects, because the levels of each mRNA were similar, with even a modest increase in tauC3 S416E message (Fig. 3H). Rather, this observation is consistent with tauC3 protein being degraded via its CHIP interaction. Together, these results indicate that the pS416

modification partially protects tauC3 from CHIP-mediated ubiquitination and turnover.

**Structural basis for phosphorylation's impact on CHIP binding**
To further understand the molecular mechanism by which tauC3 S416 phosphorylation inhibits the interaction with CHIP, we solved the X-ray

co-crystal structure of CHIP's TPR domain bound to an acetylated 10-mer peptide (Ac-SSTGSIDMVD-OH) corresponding to the tauC3 C-terminus at 1.8 Å (PDB 8FYU; Fig. 4A). Overall, we found that the orientation of the tauC3 peptide was similar to the "U-shaped" arrangement previously observed with other CHIP-binding peptides, with only a minor -1.25 Å shift of the backbone register[36,43] (Supplementary Fig. S4A). Key molecular contacts with side chains were also maintained, such as the "carboxylate clamp" interactions, involving coordination of tauC3's Asp421 side chain and carboxylate-terminus by cationic Lys30 and Lys95 side chains in CHIP's TPR domain (Fig. 4B). However, we did note a few distinct interactions, including between tauC3's Asp418 and CHIP's Lys72 (Fig. 4C). Critically, this co-structure also suggested why phosphorylation of S416 weakens binding to CHIP. Specifically, we noted that tauC3's Ser416 was in close proximity with CHIP's Asp134 (Fig. 4D). Thus, phosphorylation of tauC3's S416 would be expected to create interference at this site, likely involving both steric and electrostatic clashes. To test this idea, we mutated CHIP's Asp134 (D134A) or the adjacent Phe131 residue (F131A), to alanine and tested binding of these CHIP variants to tauC3 peptides (Fig. 4E and Supplementary Fig. S4B). Introducing the F131A mutation seemed to damage the integrity of CHIP's TPR domain because it had a lower intrinsic melting temperature (Fig. 4E), weakened binding to both WT tauC3 and tauC3 S416E peptides (Fig. 4E) and inhibited ubiquitination activity (Supplementary Fig. S4C), making this protein a poor tool for further use. Fortunately, installing D134A into CHIP was tolerated; for example, it bound normally to WT tauC3 in DSF experiments (Fig. 4E). Importantly, consistent with the design, CHIP D134A retained enhanced binding to tauC3 S416E peptide compared to WT (Fig. 4E). To independently verify this finding, we performed FP assays and confirmed that CHIP D134A bound normally to WT tauC3 ($K_i$ 5.57 ± 0.34 μM) and that it regained the ability to bind phosphorylated tauC3 peptide ($K_i$ 1.84 ± 0.23 μM) (Fig. 4F, G). Together, these results support a model in which phosphorylation at S416 weakens binding to CHIP via clashes with D134 in the TPR domain. Interestingly, the D134 residue is highly conserved in evolution (Supplementary Fig. S4D, E), suggesting that this residue could be an important "sensor" of phosphorylation.

We hypothesized that the CHIP D134A variant might partially restore ubiquitination of pS416 tauC3. To test this idea, we compared the ability of WT or D134A CHIP to ubiquitinate tauC3 or tauC3 S416E in vitro (Fig. 4H). As expected, we observed more rapid and robust ubiquitination of tauC3 S416E by D134A CHIP, compared to WT CHIP. Interestingly, we also observed that D134A CHIP had relatively more activity against WT tauC3 as well, suggesting that it might be inherently more active. Auto-ubiquitination of CHIP is known to inhibit its function[44], so we reasoned that D134A might impact this regulatory step. Indeed, we found that, while the D134A mutant CHIP was adept at ubiquitinating substrates, it had reduced autoubiquitination activity (Fig. 4H). Thus, the engineered D134A CHIP variant seems to be a useful tool for studying tauC3, overcoming even the pS416 modification.

## TauC3 co-accumulates with pS416 in AD patient brains

TauC3 is known to accumulate in AD brains and it is often used as a pathological biomarker[30]. This finding is somewhat surprising because tauC3 is an excellent substrate for CHIP[36]. Based on our results, we reasoned that this build-up might occur, in part, because tauC3 is phosphorylated at pS416—limiting its binding to CHIP. To test this idea, we measured whether the tauC3 and pS416 epitopes might co-localize in the brains of AD patients. We performed multiplexed immunofluorescence for tauC3 (C3 antibody, Invitrogen) and tau pS416 (D7U2P, Cell Signaling Technology) on fixed, post-mortem human brain sections of the hippocampal CA1/CA2 and subiculum regions from donors with increasing levels of AD pathology (Fig. 5A and Supplementary Fig. S5A). Using this approach, we observed

significant increases in both tau pS416 and tauC3 across disease progression (Fig. 5B, C) and the appearance of the two PTMs strongly correlated by Pearson's analysis ($r = 0.7681$, $p = 0.0035$; Fig. 5D). In the soma of dystrophic neurons, we often observed overlapping pathology for tauC3 and tau pS416 (see Fig. 5A—inset), suggesting these two tau PTMs could be contributing to disease when co-occurring. We also confirmed that the AD samples contained tau pathology, using the AT8 antibody (Supplementary Fig. S5B). These results support the model that tauC3 pS416 accumulates in disease. However, another possibility is that the pS416 epitope is present on other tau proteoforms, which will need to be explored in more detail. Together, these results show that pS416 correlates with AD severity in humans, supporting a model in which modification of tauC3 is one contributor to its aberrant accumulation.

## MARK2 inhibits CHIP-dependent ubiquitination of tauC3

Many tau kinases have been identified[45] and at least two of these, Ca²⁺-calmodulin-dependent protein kinase II (CaMKII) and CaMKII-like kinase microtubule affinity regulating kinase 2 (MARK2/Par-1) have been explicitly shown to phosphorylate Ser416 on FL tau[39,46,47]. However, we wanted to explore whether any of these kinases might also act on Ser416 in the context of truncated, tauC3. The CaMKII isoforms are specific to the central nervous system[48], and are not expressed in the HEK293 or *Sf9* cells that we employed. Thus, we turned our attention to MARK2 as a model. As a first step, we incubated FL tau, tauC3 or phosphomimetic tauC3 S416E with recombinant MARK2 in vitro and probed for Ser416 phosphorylation by western blot. Indeed, we observed robust phosphorylation of Ser416 in both FL and tauC3, as well as slowed in-gel mobility that is consistent with tau phosphorylation (Fig. 6A). In ubiquitination assays, the rate of CHIP-mediated ubiquitination for tauC3 was slowed following phosphorylation by MARK2 (Fig. 6B–D). It is worth noting that, in these experiments, it is likely that MARK2 acts on other Ser/Thr residues. Yet, together, these data suggest that phosphorylation of tauC3 at S416 by MARK2, and likely other kinases, is one of the contributing factors to the inhibition of CHIP-mediated ubiquitination, with important implications for the accumulation of this tau proteoform in disease.

## Discussion

Mass spectrometry studies on tau isolated from the brains of AD patients has shown that this protein is subject to extensive PTMs, including phosphorylation, proteolysis, ubiquitination, and acetylation[22,49,50]. Many of these modifications correlate with disease progression and they are being pursued as promising diagnostic biomarkers. Yet, the molecular mechanisms connecting these PTMs to tau proteostasis are often lacking[22,51,52]. For example, tau proteoforms that are phosphorylated at pT181 and pT217 are strong biomarkers for AD[53], but it is not clear whether these phosphorylation events directly impact tau interactions or its accumulation. Here, we focus on exploring the intersection between two tau PTMs that are strongly linked to AD: phosphorylation at S416 and caspase cleavage to produce tauC3. We considered this system to be a good model because recent work has identified that tauC3 binds directly to the E3 ligase CHIP[36].

Using in vitro and cellular assays, we find that phosphorylation of tauC3 at pS416 is sufficient to weaken binding to CHIP (see Fig. 2), resulting in its relative accumulation in cells (see Fig. 3C). Moreover, the appearance of pS416 and tauC3 are coincident in AD patient samples (see Fig. 5), supporting a putative role of these PTMs in tau proteostasis. Based on these collective findings and in concert with observations from the literature, we propose a model to explain the hierarchy of these tau PTMs (Fig. 6D). In this speculative model, FL tau can either be phosphorylated by MARK2 or CaMKII (or potentially other kinases), to generate phosphosites in the MTBRs[39], or GSK-3β, which generates sites in the NTDs and PRR[26]. If the GSK3β pathway

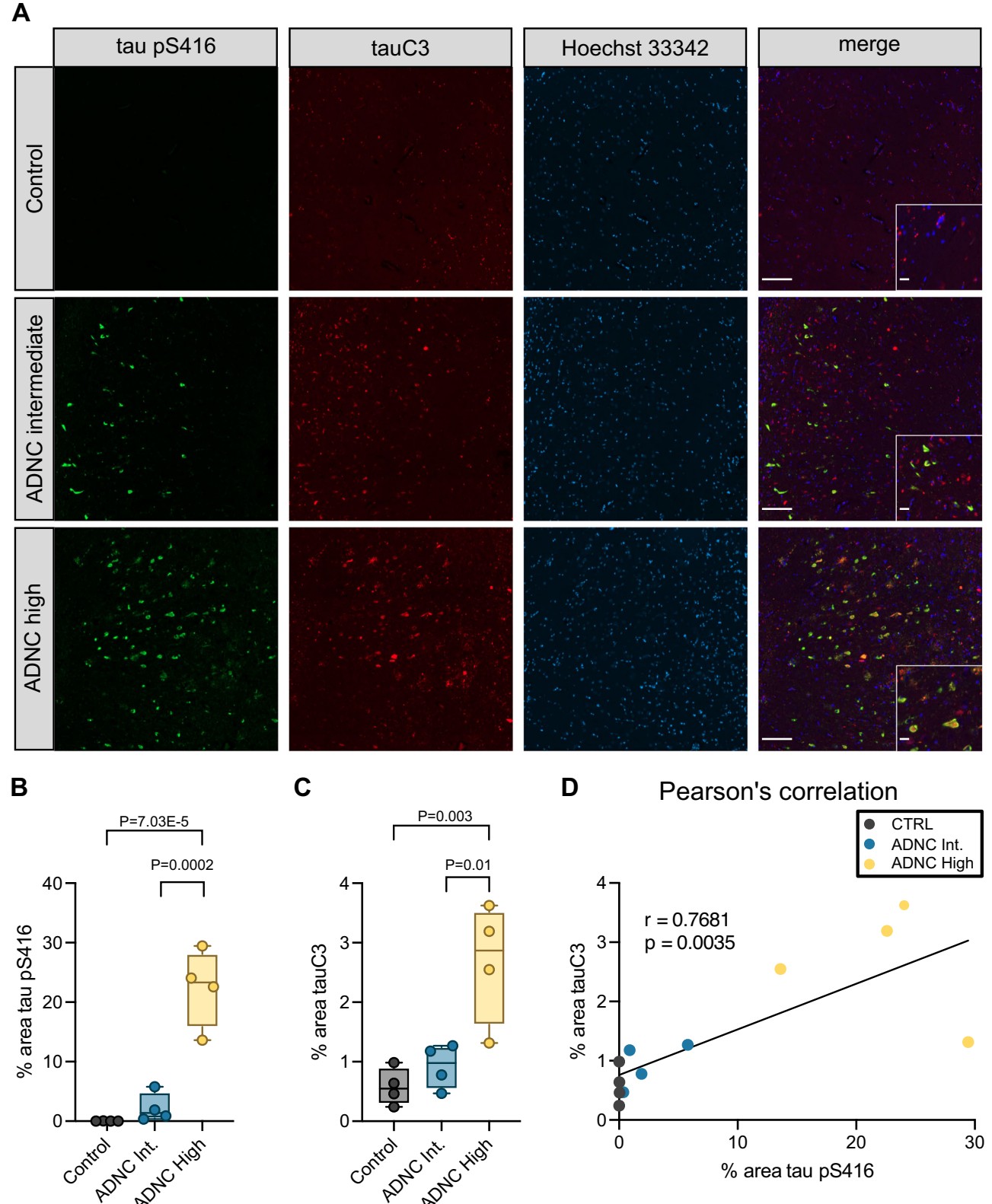

**Fig. 5 | TauC3 co-accumulates with Serine 416 phosphorylation in Alzheimer's disease patient brains. A** Representative micrographs of immunofluorescence staining of tau pS416 (green), tauC3 (red), and nuclei (Hoechst 33342, blue) from the hippocampal CA1/CA2 region of human patient brains across increasing spectrum of Alzheimer's Disease Neuropathological Change (ADNC) scoring. Scale bars = 50 μM; insets = 10 μM. **B** Quantification of tau pS416 staining across ADNC score. Box plot bounds dictate 25th–75th percentile, whiskers define sample minima and maxima, and line shows sample median. Analysis was performed on

four unique patient samples for each score. Statistical significance was determined by one-way ANOVA with Tukey's post hoc analysis. **C** Quantification of tauC3 staining across ADNC score. Box plot bounds dictate 25th–75th percentile, whiskers define sample minima and maxima, and line shows sample median. Analysis was performed on four unique patient samples for each score. Statistical significance was determined by one-way ANOVA with Tukey's post hoc analysis. **D** Two-sided Pearson's analysis showing correlation of increasing tau pS416 and tauC3 across ADNC score.

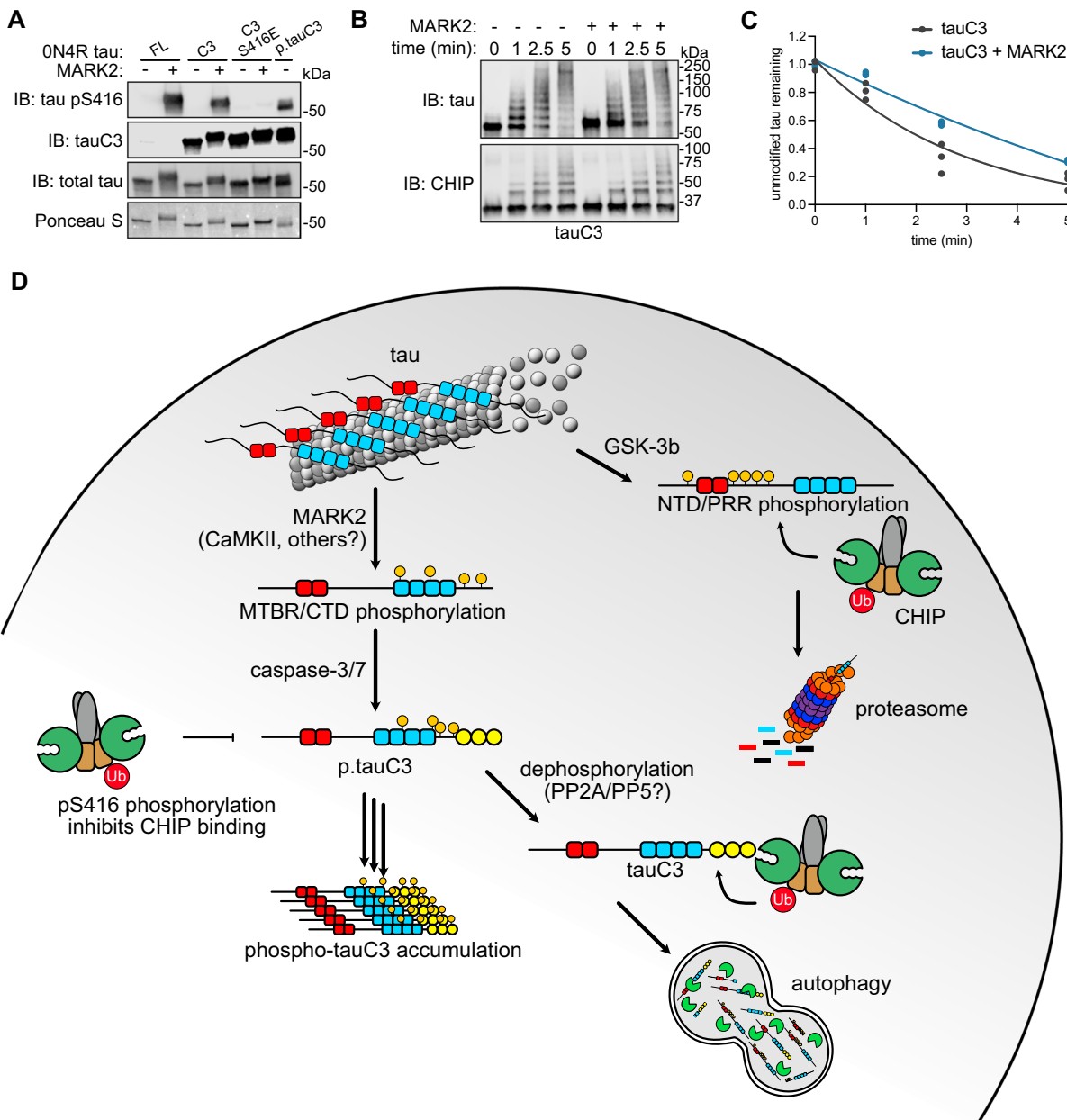

**Fig. 6 | MARK2 inhibits CHIP-dependent ubiquitination of tauC3, in part, by phosphorylating serine 416. A** In vitro phosphorylation of varying tau proteoforms by MARK2. Varying tau species were incubated overnight with recombinant MARK2, and relative phosphorylation of Ser416 was visualized by western blot. Ponceau S was used as loading control. Image is representative of duplicate experiments. **B** In vitro ubiquitination of unmodified tauC3 or MARK2-phosphorylated tauC3 by CHIP. Samples were collected at the denoted timepoints, quenched in SDS-PAGE loading buffer, and analyzed by western blot. **C** Quantification of unmodified tau remaining in (**B**). Samples were normalized to 0-min time point and curves were fit using one-phase exponential decay (*n* = 3 technical replicates). **D** Cartoon depicting the role of tau PTMs on interactions with CHIP and shuttling to various degradation pathways.

predominates, then those tau proteoforms seem likely to bind directly to CHIP and are cleared through the UPS[26]. This result is broadly consistent with our observations that okadaic acid treatment enhances binding of FL tau to CHIP (see Fig. 2). This result is also broadly consistent with recent evidence that CHIP binds FL p-tau 10-fold better than it binds to FL tau that is unmodified[23]. The exact binding sites for CHIP on FL p-tau are not yet clear, but it seems to involve an internal phospho-degron that helps tune tau levels. However, if tau is proteolyzed by caspase-3/7 to generate tauC3, then this protein binds directly to CHIP through its new, C-terminal EEVD-like motif[36] and it is then degraded by the lysosomal-autophagy pathway[54]. In addition, ubiquitination of tau might impact aggregation and other features[5,55]. This C-terminal binding affinity is ~10-fold tighter than the binding of CHIP

to the internal phospho-degron(s), so it would likely be the preferred site for tauC3, perhaps guiding a hierarchy of outcomes. Caspases are known to be active in adult neurons during non-apoptotic signaling[30], so it seems likely that tauC3 might be normally produced to feed into this degradation pathway. Here, we show that phosphorylation of tauC3 at pS416 is sufficient to partially limit degradation of tauC3. We propose that pS416 creates a requirement for dephosphorylation, likely by PP2A or PP5[56,57], the major enzymes that use tau as a substrate. It is not clear whether MARK2/CaMKII activity occurs before or after caspase cleavage. However, phosphorylation of tau at a nearby residue, S422, blocks caspase cleavage[58] and, more generally, phosphorylated tau is a poor substrate for caspases[59], which could be an important determinant of this hierarchy. Of course, these predictions

await more rigorous exploration, including in primary cells and mouse models of disease.

While this proposed model is likely to be a simplistic representation, it does serve to highlight how CHIP plays multiple, but non-overlapping, roles in tau proteostasis. For example, CHIP binds different tau proteoforms (e.g. tauC3, FL p-tau) in distinct ways and tau phosphorylation (e.g. GSK3β-mediated, pS416) can have diametrically opposite effects on CHIP's apparent affinity. Specifically, it enhances the affinity of FL tau for CHIP by 10-fold but weakens the affinity of tauC3 for CHIP. When properly balanced, we envision that CHIP uses both the GSK3β and MARK2/CaMKII pathways, along with the canonical Hsp70-mediated mechanism[5,19], to remove a wide variety of tau proteoforms (Fig. 6D). This broad capacity to identify different tau proteoforms might explain why CHIP is such a strong regulator of total tau[6]. Yet, an over-production of tauC3, combined with aberrant phosphorylation of pS416 and/or decreased phosphatase activity, might disrupt this balance and give rise to a tendency to form NFTs under specific conditions, especially if CHIP is limiting[36].

One of the goals of understanding the molecular mechanisms of tau PTMs is to identify putative drug targets and link these to specific tau proteoform biomarkers[60]. Our studies showed that MARK2 phosphorylation of tauC3 slows CHIP-dependent ubiquitination (see Fig. 6), suggesting that this kinase, and likely CAMKII[48], are important in tauC3 proteostasis. MARK2 had previously been linked to AD; for example, MARK2-mediated phosphorylation weakens the interaction of tau with microtubules and promotes tau's cytosolic accumulation[39,61] and certain MARK2 sites, such as Ser262, Ser324, and Ser396, are elevated in AD[62]. Moreover, MARK2 phosphorylation has been shown to promote tau liquid-liquid phase separation, a process that might promote the transition into amyloid fibrils[63]. Our results add to this series of observations, suggesting that kinase activity at pS416 might be particularly important for regulating tauC3.

## Methods

### Ethical statement
The de-identified, post-mortem tissues, determined to be exempt from institutional review board approval in accordance with University of California at San Francisco (UCSF) policy, were sourced from the Neurodegenerative Disease Brain Bank (https://memory.ucsf.edu/neurodegenerative-disease-brain-bank).

### Cell lines and culture conditions
HEK 293T cells (ATCC) were cultured in Dulbecco's Modified Eagle Medium (DMEM) supplemented with 10% Fetal Bovine Serum (FBS) and penicillin/streptomycin. Parent HEK 293 Flp-In T-REx cells (Thermo Fisher) were cultured in 10% FBS-DMEM supplemented with 1X penicillin/streptomycin, 15 μg/mL blasticidin, and 100 μg/mL zeocin. All mammalian cells were maintained at 37 °C / 5% CO$_2$ in a humidified incubator.

Construction of the stable dox-inducible HEK 293 eGFP-tau reporter cell lines was performed as described[36]. In brief, parent HEK 293 Flp-In T-REx cells were harvested by trypsinization, resuspended in electroporation buffer containing 5 μg eGFP-tau plasmid DNA and 45 μg pOG44 plasmid DNA. Cells were electroporated using program Q-001 on a Lonza 4D-Nucleofector. Electroporated cells were immediately diluted in 10% FBS-DMEM and cultured for 48 h to recover. Following recovery, cells were selected with 100 μg/mL hygromycin until single colonies arose. Colonies were subsequently picked and screened for dox-inducible expression of eGFP-tau by fluorescence microscopy (Echo Revolve) and western blotting.

### NanoBiT live cell PPI assay
CHIP-substrate NanoBiT PPI assay was performed as previously described[42]. Briefly, HEK293T cells were seeded in poly-D-lysine coated 96-well plates (Corning, flat bottom, white opaque) in Opti-MEM. Cells were transfected with 50 ng each SmBiT/LgBiT DNA using Lipofectamine 3000 (Thermo Fisher) in Opti-MEM according to the manufacturer's instructions. Transfections were performed for 24 h, after which the media was replaced with fresh Opti-MEM containing DMSO or 30 nM okadaic acid. Cells were incubated for an additional 16 h at 37 °C/5% CO$_2$ in a humidified incubator. Following this incubation, 12.5 uL 1X Nano-Glo live cell luciferase reagent (Promega) diluted in LCS buffer (Promega) was added to each well, and samples were incubated for 10 min at room temperature in the dark to allow luminescence levels to stabilize. Luminescence recordings were performed on a SpectraMax M5 plate reader with 500 ms integration time in well-scan mode acquiring 9 readings per well. Un-transfected cells were used to obtain background measurements which were subtracted from observed luminescence values.

### Protein purification
Recombinant human CHIP was expressed from a pMCSG7 construct with N-terminal tobacco etch virus (TEV)−cleavable 6His-tag. pMCSG7-CHIP was transformed into BL21DE3 (New England Biolabs) E. coli and grown in terrific broth (TB) to OD600 = 0.5 at 37 °C. Cells were cooled to 16 °C, induced with 500 μM isopropyl β-D-1-thiogalactopyranoside (IPTG), and grown overnight. Cells were collected by centrifugation (-6000 × g), resuspended in binding buffer (50 mM Tris pH 8.0, 10 mM imidazole, 500 mM NaCl) supplemented with cOmplete protease inhibitors (Roche), and sonicated. The resulting lysate was clarified by centrifugation and the supernatant was applied to Ni$^{2+}$-NTA His-Bind Resin (Novagen). Resin was washed with binding buffer and His wash buffer (50 mM Tris pH 8.0, 30 mM imidazole, 300 mM NaCl), and then eluted from the resin in His elution buffer (50 mM Tris pH 8.0, 300 mM imidazole, 300 mM NaCl). Following this step, the N-terminal His tag was cleaved by overnight dialysis with TEV protease at 4 °C and purified by size exclusion chromatography (SEC) (HiLoad Superdex-200 16/600 column, GE Healthcare) in CHIP storage buffer (50 mM HEPES, 10 mM NaCl, pH 7.4).

CHIP TPR Domain (human, AA 22−154) was expressed from a pMCSG7 construct with an N-terminal TEV-cleavable 6xHis tag, as previously described[36]. E. coli were grown in TB at 37 °C, induced with 1 mM IPTG in log phase, cooled to 16 °C and grown overnight. Ni-NTA purification and tag removal were conducted as for full-length CHIP. Protein was further purified on a Mono S cation exchange column (GE Healthcare) and stored in CHIP TPR storage buffer (10 mM Tris, 150 mM NaCl, 2 mM DTT, pH 8.0).

Recombinant, unmodified 0N4R tau proteins were expressed from a pMCSG7 construct with N-terminal TEV − cleavable 6His-tag. pMCSG7-tau was transformed into BL21DE3 E. coli and grown in terrific broth (TB) to OD600 = 0.5 at 37 °C. Betaine (10 mM) and NaCl (500 mM) were added to media and tau was induced by addition of IPTG (200 μM) for 3 h at 30 °C. Cells were collected by centrifugation (-6000 × g), resuspended in tau lysis buffer (1X D-PBS, 2 mM MgCl$_2$, 1 mM DTT, 1 mM EDTA, pH 7.4) supplemented with cOmplete protease inhibitor and 1 mM phenylmethylsulfonyl fluoride (PMSF), and lysed by sonication. The lysate was clarified by centrifugation and applied to cOmplete His-tag purification resin (Sigma Aldrich) for 1 h at 4 °C with rotation. The resin was washed with tau lysis buffer and bound protein was eluted with tau elution buffer (1X D-PBS, 300 mM imidazole, 200 mM NaCl, 5 mM β-mercaptoethanol, pH 7.4). His-tags were removed by addition of TEV protease and overnight dialysis into tau buffer (1X D-PBS, 2 mM MgCl$_2$, 1 mM DTT, pH 7.4) at 4 °C. Proteins were subsequently concentrated and purified by reverse-phase HPLC as previously described[64]. Solvent was removed by lyophilization, and the resulting protein samples were resuspended in tau buffer.

Phosphorylated tauC3 was expressed in Sf9 insect cells from a 438B pFastBac vector containing an engineered 6xHis tag and a TEV protease cleavage site (Addgene). Following two rounds of virus amplification, the protein was expressed in Sf9 cells by infecting with

recombinant baculovirus at 20 to 40 µL virus per million cells in culture flasks and incubated for 3 days at 27 °C with shaking at 120 RPM. Cells are then collected by centrifugation (~6000 × *g*) for 20 min at 4 °C and stored at −80 °C. The cells are resuspended in Ni lysis buffer (20 mM Tris, pH 8.0, 500 mM KCl, 10 mM imidazole, 10% glycerol) supplemented with EDTA-free protease inhibitor cocktail (Roche) and 6 mM beta-mercaptoethanol. Cells were lysed in a Dounce homogenizer, and the lysates were boiled for 20 min to denature and precipitate nearly all proteins except for tau. The lysate was centrifuged (80000 RCF) at 4 °C for 30 min to clarify and the supernatant was incubated with HisPur Ni-NTA resin (Thermo Scientific) at 4 °C for 1 h. The resins were then washed and eluted with elution buffer (20 mM Tris-HCl, 100 mM KCl, 6 mM β-mercaptoethanol, 300 mM imidazole, pH 8.0). The fractions containing tau, as judged by SDS-PAGE, were dialyzed with a dialysis buffer (50 mM Tris-HCl, 100 mM KCl, 6 mM β-mercaptoethanol, 5% glycerol, pH 8.0) containing TEV protease to cleave the His tag at 4 °C overnight. The dialyzed proteins were concentrated and purified by SEC (HiLoad Superdex-200 16/600 column, GE Healthcare) into tau buffer. Phosphorylation was confirmed by gel-shift and western blotting with phospho-specific antibodies.

MARK2 phosphorylation of recombinant tau proteins was performed as previously described[39] with slight modification. In brief, recombinant GST-tagged MARK2 (Promega) was incubated with tau at 30 °C at 1:100 ratio of MARK2:tau in phosphorylation buffer (25 mM PIPES, 100 mM NaCl, 5 mM MgCl2, 2 mM EGTA, 1 mM DTT, 1 mM benzamidine, 0.5 mM PMSF, 1 mM ATP, pH 6.8) for 18 h. Following, samples were incubated with equilibrated Pierce glutathione resin (Thermo Fisher) for 1 h to remove MARK2, followed by desalting and buffer exchange into tau buffer over Zeba protein desalting columns (Thermo Fisher).

### Peptide synthesis
Peptides were synthesized by Fmoc solid phase peptide synthesis on a Syro II peptide synthesizer (Biotage) at ambient temperature and atmosphere on a 12.5 µmol scale using pre-loaded Wang resin. Coupling reactions were conducted with 4.9 eq of HCTU (O-(1H-6-chlorobenzo-triazole-1-yl)−1,1,3,3-tetramethyluronium hexafluoro-phosphate), 5 eq of Fmoc-AA-OH and 20 eq of N-methylmorpholine (NMM) in 500 µL of N,N dimethyl formamide (DMF). Reactions were run for 8 min while shaking. Each position was double coupled. Fmoc deprotection was conducted with 500 µL 40% 4-methypiperadine in DMF for 3 min, followed by 500 µL 20% 4-methypiperadine in DMF for 10 min, and six washes with 500 µL of DMF for 3 min. Acetylation was achieved by reaction with 20 eq acetic anhydride and 20 eq NMM in 500 µL DMF for 1 h while shaking. Peptides were cleaved with 500 µL of cleavage solution (95% trifluoroacetic acid 2.5% Water 2.5% triisopropylsilane) while shaking for 1 h. Crudes were precipitated in 10 mL cold 1:1 diethyl ether: hexanes. Peptide crudes were solubilized in a 1:1:1 mixture DMSO: water: acetonitrile and purified by HPLC on an Agilent Pursuit 5 C18 column (5 mm bead size, 150 mm × 21.2 mm) using an Agilent PrepStar 218 series preparative HPLC. The mobile phase consisted of A: Water 0.1% Trifluoroacetic acid and B: Acetonitrile 0.1% Trifluoroacetic acetic acid. Solvent was removed under reduced atmosphere and 10 mM DMSO stocks were made based on the gross peptide mass. Purity was confirmed by LC/MS. Stocks were stored at −20 °C. Fluorescence polarization tracer peptides and tau phospho-peptides were synthesized by Genscript.

### Western blotting
Prepared samples were separated on precast 4–20% SDS-PAGE gradient gels (Bio-Rad) for 35 min at 200 V. Proteins were transferred to nitrocellulose membranes using a Trans-Blot Turbo Transfer System (Bio-Rad). Membranes were blocked in Odyssey TBS Blocking Buffer (Licor) for 1 h at room temperature and then incubated in primary antibody in 1X Tris-Buffer saline containing 0.05% Tween-20 (1X TBS-T), plus 5% non-fat dry milk powder, and 0.02% NaN3 overnight at 4 °C with rotation. The following day, membranes were washed 3 times in 1X TBS-T and then incubated in secondary antibody diluted 1:10,000 in Odyssey TBS Antibody Diluent (Licor) for 1 h at room temperature. Following incubation with secondary antibody, membranes were washed 3 times in 1X TBS-T and imaged on an Odyssey Fc Imaging System (Licor). Quantification was performed by densitometry analysis in ImageJ (NIH). Raw images, along with select quantifications, can be found in Fig. S6. Antibodies and dilutions used were as follows: CHIP (1:2000, Abcam #ab109103), total tau D-8 (1:1000, Santa Cruz Biotech #sc-1661060), tau pS416 (1:1000, Cell Signaling, 15013), tauC3 (1:1000, Millipore, MAB5430), tau pS202/pT205 AT8 (1:1000, Thermo Fisher, #MN1020), tau pS396 (1:1000, Santa Cruz Biotech, #sc-32275), IRDye 680RD goat anti-mouse secondary (1:10,000, LI-COR #926-68070), IRDye 800CW goat anti-rabbit secondary (1:10,000, LI-COR #926-32211).

### CHIP-tau binding ELISA
Purified CHIP (1 µM) or buffer-matched control was immobilized in 96-well plates (Fisher Scientific, non-sterile, clear, flat-bottom) in CHIP buffer (50 mM HEPES, 10 mM NaCl, pH = 7.4) overnight at 37 °C. The protein sample was removed, and wells were washed 3X with phosphate-buffered saline with 0.05% Tween-20 (PBS-T) for 3 min with rotation at room temperature. Tau samples were prepared as a 3-fold dilution series in tau binding buffer (25 mM HEPES, 40 mM KCl, 8 mM MgCl2, 100 mM NaCl, 0.01% Tween, 1 mM DTT, pH 7.4) and incubated at RT for 3 h with rotation. Tau was removed, and wells were washed as described. Samples were blocked in 5 % non-fat dry milk in tris-buffered saline (TBS), and then incubated with primary anti-tau (Santa Cruz Biotech, 1:2000) followed by HRP-conjugated secondary antibody (Anaspec, 1:2000). Antibodies were dissolved in 1X TBS with 0.05% Tween-20. Incubations were performed for 1 h at RT with rotation, separated by wash steps, as described[23]. TMB substrate (Thermo Fisher) was then added to the wells and incubated for 15 min at RT, followed by quenching with 1 M HCl. Absorbance readings were performed on a SpectraMax M5 plate reader at $OD_{450}$. The data was background subtracted to buffer only controls, normalized to maximal binding, and binding curves were fit using non-linear regression in Prism 9.0 (GraphPad).

### In vitro ubiquitination assays
In preparation for in vitro ubiquitination reactions, four 4× stock solutions were prepared containing (1) Ube1 + UbcH5c (400 nM Ube1 and 4 µM UbcH5c), (2) Ubiquitin (1 mM Ub), (3) CHIP + substrate (4 µM CHIP and 4 µM substrate) and (4) ATP + MgCl2 (10 mM ATP and 10 mM MgCl2) in ubiquitination assay buffer (50 mM Tris pH 8.0, 50 mM KCl). Ubiquitination reactions were generated by adding 10 µL of each 4× stock, in order from 1 to 4, for a final volume of 40 µL (100 nM Ube1, 1 µM UbcH5c, 250 µM ubiquitin, 2.5 mM ATP, 2.5 mM MgCl2, 1 µM CHIP and 1 µM substrate). Reactions were then incubated at room temperature, and 10 µL aliquots were collected at each time point and quenched in 5 µL 3× SDS−PAGE loading buffer. Samples were separated by SDS-PAGE and analyzed by western blotting. Quantification of unmodified tau remaining was performed by densitometry analysis in ImageJ (NIH).

### Mapping of tau PTMs by liquid chromatography with tandem mass spectrometry
A sample of protein (8 µg) was denatured and reduced with 8 M urea in 100 mM ammonium bicarbonate (pH 8) buffer and 100 mM dithiothreitol (DTT) at 60 °C for 30 min, followed by alkylation with 100 mM iodoacetamide at room temperature in the dark for 1 h. The sample was then incubated 4 h with trypsin (1:20 weight/weight) at 37 °C. The peptides formed from the digestion were further purified by C18 ZipTips (Millipore) and analyzed by LC-MS/MS. The MS/MS analyses were conducted using either an Q Exactive Plus Orbitrap (QE) or a

Fusion Lumos Orbitrap (Lumos) mass spectrometer (Thermo Scientific). Higher-energy collisional dissociation was used to produce fragmented peptides. The mass resolution of precursor ions was 70000 on the QE and 120000 on the Lumos. The mass resolution of fragment ions was 17500 on the QE and 30000 on the Lumos, respectively. The LC separation was carried out on a NanoAcquity UPLC system (Waters) for both the QE and the Lumos. The LC linear gradient on the QE was increased from 2–25% B (0.1% formic acid in acetonitrile) over 48 mins followed by 25–37% B over 6 min and then 37–40% B over 3 mins at a flow rate of 400 nL/min. The LC linear gradient on the Lumos was increased from 2–5% B over 3 mins followed by 5–30% B over 72 mins and then 30–50% B over 2 mins at a flow rate of 300 nL/min. The acquired MS/MS raw data was converted into peak lists using an in-house software PAVA and then analyzed using Protein Prospector search engine. The Max. missed cleavages was set to 2. The precursor / fragment mass tolerances were set at 20 ppm / 20 ppm for the QE and 10 ppm / 20 ppm for the Lumos. The maximum false discovery rate (FDR) was set at 1% for both protein and peptide levels. The Threshold of the Site Location in Peptide (SLIP) score was set at 6. For all peptides, phosphorylation modification at serine, threonine, and tyrosine residues was selected. Visualization of tau PTMs was performed using Protter[65]. Results are deposited in PRIDE (see Data Availability) and peptide sequences can be found in Supplementary Data S1.

## Differential scanning fluorimetry

DSF was performed with a 10 μL assay volume in 384-well Axygen quantitative PCR plates (Fisher Sci) on a qTower[3] real-time PCR thermal cycler (Analytik Jena). Fluorescence intensity readings were taken over 70 cycles in "up-down" mode, where reactions were heated to desired temp and then cooled to 25 °C before reading. Temperature was increased 1 °C per cycle. Each well contained 5 μM CHIP, 5× Sypro Orange dye (Thermo Fisher), and varying concentrations of peptide in DSF assay buffer (25 mM HEPES pH 7.4, 50 mM KCl, 1 mM TCEP, 0.2% CHAPS, 1%DMSO). Fluorescence intensity data was truncated between 30–60 °C, plotted relative to temperature, and fit to a Boltzmann Sigmoid in GraphPad Prism 9.0. CHIP apparent melting temp (Tm$_{app}$) was calculated based on the following equation:

$$Y = \text{Bottom} + ((\text{Top} - \text{Bottom})/(1 + \exp(Tm - T/\text{Slope}))) \tag{1}$$

## Fluorescence polarization

Fluorescence polarization (FP) assays were performed in 18 μL in a Corning black 384 well round bottom low volume plate and measurements made on a SpectraMax M5 multimode plate reader at 21 °C. A 2× stock of CHIP + Tracer was made in CHIP FP assay buffer, so that the final assay concentration of CHIP was 1.58 μM and tracer was 20 nM. The 2× peptide competitor stocks were prepared in CHIP FP Dilution buffer (25 mM HEPES pH 7.4, 50 mM KCl 0.01% Triton X-100, 2% DMSO) in three-fold dilutions. 2× CHIP + Tracer and peptide competitor solutions were mixed at equal volumes and incubated at 25 °C in the dark for 15 min. Raw polarization (mP) values were background subtracted to tracer alone and plotted relative to log$_{10}$ (competitor). Data was fit to the model for [inhibitor] versus response (three parameters) in Graphpad Prism 9.0. IC$_{50}$ values were calculated based on the equation:

$$Y = \text{Bottom} + (\text{Top} - \text{Bottom})/(1 + (X/\text{IC50})) \tag{2}$$

$K_i$ values were calculated as previously described[66] using the equation:

$$K_i = [I]_{50}/([L]_{50}/K_d + [P]_0/K_d + 1) \tag{3}$$

## Co-immunoprecipitation assays

Cells were washed once in 1X Dulbecco's Phosphate Buffered Saline (D-PBS) and lysed in ice cold Pierce Immunoprecipitation (IP) lysis buffer (25 mM Tris-HCl pH 7.4, 150 mM NaCl, 1 mM EDTA, 1% NP-40). Lysates were collected by scraping, transferred to 1.5 mL microcentrifuge tubes, and incubated on ice for 15 min. Following, lysates were clarified centrifugation at 14,000 RCF for 10 min at 4 °C and supernatants were transferred to new microcentrifuge tubes. Relative protein concentrations were determined by bicinchoninic acid assay (Pierce), and samples were normalized to the lowest protein concentration. A representative input sample of 40 μL was collected, mixed with 20 μL 3X SDS-PAGE loading buffer (188 mM Tris-HCl pH 6.8, 3% SDS, 30% glycerol, 0.01% bromophenol blue, 15% β-mercaptoethanol), denatured at 95 °C for 5 min, and stored at 4 °C for later analysis.

For tau immunoprecipitations, an aliquot (30 μL) of Protein A/G PLUS-agarose (Santa Cruz Biotech) per condition was collected and washed twice in ice cold IP lysis buffer. Then, an aliquot (30 μL) of resuspended beads and anti-Tau-5 antibody (1:250, Thermo Fisher, AHB0042) were added to prepared lysates and incubated at 4 °C overnight with rotation. The following day, beads were collected by centrifugation and supernatants were removed. Beads were washed 3 times in ice cold IP lysis buffer, and bound proteins were eluted by addition of 60 μL 1X SDS-PAGE loading buffer and denaturing at 95 °C for 5 min. Inputs and eluates were subsequently analyzed by SDS-PAGE followed by western blotting. Quantification was performed by densitometry analysis in ImageJ (NIH).

## qPCR

Relative quantitation of *MAPT* gene expression was performed as previously described[67]. Briefly, cells were collected by centrifugation and RNA was extracted via Quick-RNA Miniprep Kit (Zymo Research) and complementary DNA (cDNA) was synthesized via SensiFast cDNA Synthesis Kit (Meridian Bioscience), both according to the manufacturer's instructions. Samples were prepared for qPCR in both technical and biological triplicates in 5-μl final volumes using SensiFAST SYBR Lo-ROX 2X Master Mix (Meridian Bioscience) with qPCR primers (Integrated DNA Technologies) at a final concentration of 0.2 μM and cDNA diluted at 1:3. qPCR was performed on a QuantStudio 6 Pro Real-Time PCR System (Applied Biosystems) using QuantStudio Real Time PCR software (v.1.3) with the following Fast 2-Step protocol: (1) 95 °C for 20 s; (2) 95 °C for 5 s (denaturation); (3) 60 °C for 20 s (annealing/extension); (4) repeat steps 2 and 3 for a total of 40 cycles; (5) 95 °C for 1 s; (6) ramp 1.92 °C s$^{-1}$ from 60 °C to 95 °C to establish melting curve. Expression fold changes were calculated using the 2^-ΔΔCt method and normalized to housekeeping gene *ACTB*.

## X-ray crystallography

The protein solution was prepared by mixing a 1:2 molar ratio of human CHIP-TPR, at 6 mg/ml, and the 6-mer or 10-mer tau peptide in protein buffer (10 mM Tris-HCl pH 8.0, 150 mM NaCl and 2 mM DTT), and incubated on ice for 30 min. Crystals of the complex were grown at room temperature by hanging-drop by mixing 100 nL of the protein solution with 100 μL of the crystallization condition (0.1 M CaCl$_2$, 0.1 M HEPES (pH 7.4), 28% PEG 4 K (10mer) or 25% PEG 3350 (6mer)) by Mosquito Nanoliter Dropsetter (TTPLabtech). Crystals appeared within 48 h and were harvested ~ 1 week after setup by flash-freezing in liquid nitrogen using a cryogenic solution of 50% MPD in the crystallization condition. Data were collected at Lawrence Berkeley National Laboratory Advanced Light Source beamline 8.3.1. Diffraction images were processed using Xia2 with the Dials pipeline[68]. Automatic molecular replacement was performed using the online Balbes tool[69]. The resulting structure models were refined over multiple rounds of restrained refinement and isotropic B-factor minimization with

Phenix[70]. Structural visualization and rendering for figures were performed using UCSF ChimeraX-1.4. See Supplementary Methods for additional information.

## Histopathology

Samples for immunofluorescence were formalin fixed paraffin embedded and cut at 8 μm. Slides were deparaffinized and subjected to hydrolytic autoclaving at 121 °C for 10 min in citrate buffer (Sigma, C9999). Following blocking with 10% normal goat serum (Vector laboratories, S-1000), sections were incubated with primary antibodies, tau pS416 (Cell Signaling, 15013) 1:200, tauC3 (Millipore, MAB5430) 1:250 or AT8 (Thermo Fisher MN1020) 1:250, overnight at room temp. After washing, sections were incubated in secondary antibodies Alexa Fluor goat anti rabbit 488 (Thermo Fisher A11008) and Alexa Fluor goat anti mouse 647 (Thermo Fisher A21235) both 1:500 for 120 min at room temp. Sections were then washed and incubated in Hoechst (Life Technologies H3570) 1:5000 for 10 min and then rinsed with DI water and coverslipped using Permafluor aqueous mounting medium (Thermo Scientific TA030FM). Slides were imaged using the Zeiss AxioScan.Z1. Digital images were analyzed using the Zeiss Zen 3.5 (blue edition) Analysis software. To quantify phospho-tau (Ser416) and cleaved tau (tauC3) neuropathology, a pixel intensity threshold was determined using a slide from a clinically diagnosed AD patient with high neuropathologic change and was then applied to all slides. Regions of interest were drawn in the subiculum and the CA1/CA2 of the hippocampus, and the percent area of pixels positive for staining in each region was determined.

## Reporting summary

Further information on research design is available in the Nature Portfolio Reporting Summary linked to this article.

## Data availability

Crystallography data are deposited in the Protein Data Base (8FYU [https://www.rcsb.org/structure/8FYU]). The mass spectrometry proteomics data have been deposited to the ProteomeXchange Consortium via the PRIDE partner repository with the dataset identifier, PXD052540 [https://www.ebi.ac.uk/pride/review-dataset/8f9401e858354d818c1631cea577c203], and peptide sequences are in Supplementary Data S1. Raw western blots are available in Supplementary Fig. S6. A list of primers and all other data is available in the Source Data document. Source data are provided with this paper.

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

## Acknowledgements

This work was supported by grants from the Alzheimer's Association (to C.N.) and the NIH (AG077842 to M.D.C. and AG068125 to D.R.S., C.S.C. and J.E.G.). Additional support was provided by the BrightFocus Foundation and the Tau Consortium (J.E.G.). Human tissue samples were provided by the Neurodegenerative Disease Brain Bank at the University of California, San Francisco, which receives funding support from NIH grants P01AG019724 and P50AG023501, the Consortium for Frontotemporal Dementia Research, and the Tau Consortium.

## Author contributions

C.M.N. and J.E.G. designed the studies and wrote the manuscript. All authors edited the manuscript. C.M.N., S.P., A.C.T. and M.D.C. conducted biochemical and cell-based experiments, performed data analysis, and generated necessary reagents. K.B. and K.W. performed the crystallography experiments, including data collection and data processing. A.O. performed the histopathology studies. J.E.G., D.A.M., C.S.C. and D.R.S. provided project oversite, data interpretation and funding.

## Competing interests

The authors declare no competing interests.
