## [Peer Review File · Nature Communications]

Reviewers' Comments:

Reviewer #1:

Remarks to the Author:

The manuscript by Nadel et al. considers the effect of a single phosphorylation of a Tau fragment, TauC3, on its interaction with CHIP E3 protein ligase. They provide strong evidence that phosphorylation of S416 decreases the interaction and the ubiquitination of tauC3. This observation is of interest as the TauC3 fragment is known to accumulate in AD. The structure of the complex between CHIP and a peptide corresponding to TauC3 C-terminus is provided. Based on this structure, a CHIP mutation is designed and able to restore to some extent the binding of the phosphorylated fragment. The study is relevant as tau degradation is complex, including via the CHIP pathway that is mediated by the chaperone HSP70 but can also act without the need for the chaperone, depending of the tau isoform/fragment. Most importantly, the study reveals the complexity of phosphorylation as multiple tau phosphorylation can increase tau ubiquitination mediated by Hsp70.CHIP while a single phosphorylation of TauC3 decreases the ubiquitination mediated by CHIP alone. Identification of MARK2 as the kinase that phosphorylates S416 might be a simplification as they are probably other kinases able to phosphorylate this residue and that could be made clear. While the majority of the experiments have been performed in vitro, co-localization immunofluorescence experiments in human brain slices suggest that tauC3 is phosphorylated on S416 in AD patients.

The manuscript is well written and results are presented in a logical manner. Methods are detailed.

A few specific minor remarks :

“we first confirmed that CHIP binds approximately 5-fold better to tauC3 over full length (FL) tau under basal conditions” : “binds 5-fold better” is not an appropriate characterization of the results as the dissociation constants have not been determined in this experiment – the signal is increased 5 times, the amount of complex is increased

Typo, supplementary figure 4 E : as derived from...missing information

tauC3p416 peptide would be of interest in experiments/graphs of Fig 2D-Fig S2A to facilitate comparison with the pseudo-phospho peptides

Fig 3A-B and Fig 1 E : a graphical representation for experiment 1E as in 3B would facilitate comparison of the 2 experiments/figures

Reviewer #2:

Remarks to the Author:

In this study, the author reported that phosphorylation of tauC3 at a single residue, pS416, is sufficient to block its interaction with CHIP using many biochemical assay, crystal structure, mass spectrometry analysis, and imaging in clinical samples. This is an interesting and well-presented study. However, there were some defects and deficiencies in this study.

1. At first, the author hypothesized that HSP70-like mechanism might govern the binding of tauC3's C-terminal degron to CHIP. Why didn't the author compare HSP70 and Tau3C as control? In addition, the author should use the C-terminal peptide (phosphor-mimic 10mers) of HSP70 as control in Fig 2D-I. Finally, additional structural modeling of C-terminal peptide of HSP70 was necessary for comparison between HSP70 and Tau3C.

2. The author treated with the protein phosphatase inhibitor, okadaic acid (30 nM). However, there is no direct evidence that okadaic acid prevents dephosphorylation of Tau3C, especially S416. The references cited in this manuscript showed selective inhibition of PP2A by okadaic acid induced an AD-like hyperphosphorylation and accumulation of tau. Many of the sites that are dephosphorylated by PP2A are phosphorylated by either GSK-3 β or Cdk5. These include sites Ser199, Ser202, Thr205, Ser396, and Ser404. The author must show direct evidence of okadaic acid mediated dephosphorylation in Tau3C using mass spectrometry or western blot assay.

3. The author performed mass spectrometry analysis for the identification of phosphorylation and ubiquitylation of tau isoforms. However, no mass spec data provided. The author should provide the detailed information of identification of phosphorylation and ubiquitylation including database search score, FDR level, localization score for PTM sites, etc. In addition, mass spectrometry raw data should be deposited in public repository.

4. In Fig 5, the author observed that significant increases in both tau pS416 and tauC3 across disease progression. However, high correlation between pS416 and tauC3 also suggests the possibility that phosphorylation may simply increase due to an increase in the amount of Tau3C protein. Did the author check the amount of FL tau using immunofluorescence?

5. The author suggested that MARK2-mediated phosphorylation of tauC3 at pS416 is sufficient to block binding to CHIP using in vitro and cellular assays. However, the author performed only in-vitro assay using recombinant MARK2 and tau isoforms. The reference paper cited by author showed that multiple sites are phosphorylated as well as S416. It is difficult to confirm a direct relationship between MARK2 and S416 based on these results alone. At a minimum, results from in vivo assay (MARK2 knockout, etc.) are required. In addition, If the author was going to do an in vitro assay, why didn't the author experiment with CAMKII?

6. The author reported that CHIP binds p.tau ~10-fold better than it binds to unmodified tau. These findings show that TPR domain of CHIP recognizes a phosphorylation-dependent degron on tau centered on pS214 and pT212. The author should discuss differences between this study and the author's recent report.

Reviewer #3:

Remarks to the Author:

In this manuscript Nadel et al provide a beautiful characterization of the effect of phosphorylation of tau3C isoform on its interaction with CHIP E3 ligase, and its subsequent cellular ubiquitination and degradation.

Through an elegant combination of structural biology, biophysics and cellular assays, the authors show that a single phosphorylation in tau3C is sufficient to inhibit the CHIP-tau3C interaction, thus providing a mechanistic explanation as to why these species accumulate in the brains of AD patients.

The manuscript is technically excellent and was a pleasure to read. I have but a few minor comments -

1) The authors state on page 3 that phosphorylation masks CHIP ubiquitination sites on tau3C, however supplementary figure 1c does not show any sites that were found to be ubiquitinated only in tau3C and not p.tau3C. If such sites were indeed identified, they should be clearly shown in that figure, otherwise the statement needs to be rephrased.

2) While the evidence that phosphorylation of tau3C weakens binding to CHIP is very clearly shown, the hindered ubiquitination ability is less so. While the ubiquitination is reduced, it still is present at a significantly higher degree for p.tau3C or tau3C S416E mutant compared to the regular tau. Is this just a matter of CHIP's affinity for tau? If this is the case, no ubiquitination should be observed for tau3C lacking the 6-7 most C-terminal residues.

3) The authors imply that the C-terminal region of tau3C is its sole binding site to CHIP. However, phosphorylation of this region only slightly reduces the affinity, and p.tau3C still binds CHIP significantly stronger than the regular tau isoform. Can the authors rule out the presence of other binding sites on CHIP for tau3C?

4) The effect of the phosphorylation on the cellular degradation of tau3C is very minor. Would this effect be better seen by CHX chase assays, for example?

Reviewer #4:

Remarks to the Author:

The quality of the crystallographic structures presented in this manuscript is good overall. It is possible that the data have been processed a little too conservatively - a CC(1/2) value of 0.57 in the highest resolution shell is very high - but processing pipelines such as xia2 tend towards the conservative by default and this is not a serious flaw with the data.

The quality of the models presented is good and the figures are clear.

Reviewer #5:

Remarks to the Author:

This manuscript by Nadel et al. describes the interaction between phosphorylation and ubiquitination of cleaved tau as it relates to the E3 ubiquitin ligase, CHIP. The authors provide biochemical, biophysical, and structural data to support their conclusions. Overall, the data in this manuscript are rigorous and provide new mechanistic insight into the competition of tau posttranslational modifications that regulate its accumulation or degradation, respectively. However, there are some claims made that should be softened and some additional experiments and discussion that would further strengthen the manuscript.

1. It is unclear how much of an impact this competition over phosphorylation/ubiquitination has on tau. How does this competition affect tau aggregation, degradation, or other outcomes?
2. Does ubiquitination by CHIP inversely block the phosphorylation of tau at S416?
3. While the S416E did significantly affect CHIP binding, the binding was fully blocked. The authors should at least discuss what other factors may be contributing to the interaction outside of this one phosphorylation site.
4. Recombinant tau is challenging to isolate without having some Hsp70 still associated. What is the purity of tau? Is there Hsp70 present from the protein production? Does this Hsp70 have an impact on the effects of CHIP on tau?
5. In a prior manuscript by this group in 2018 in NSMB, it was demonstrated that CHIP interacts most with 3R tau. Are the effects on 3R tau in line with the 4R tau?
6. The data from human brain tissue is on a limited set of patients. It may be more convincing to include CHIP and ubiquitin co-staining.
7. Does the ubiquitination of tau that reacts with S416 antibody change?
8. All western blots should include quantification.
9. In addition to regulating tau turnover, it has been shown that ubiquitination can help

promote the fibrilization of tau. The authors should discuss this in the context of their findings.

10. Limitations of the work should be included.

11. Some bold claims from the data, which shows partial blocks/reductions should be softened to better represent the data.

12. There appears to be an error in Figure 6 figure legend in reference to the quant of panel C.

13. All raw, uncut blots should be shown.

We are pleased that the five reviewers were all enthusiastic about this work and thank them for the helpful feedback. Here, we reproduce the original comments (black) and provide a point-by-point summary of the changes made in the revision (blue). These revisions include a number of new experiments that support, confirm and extend the original conclusions.

Reviewer 1.

The manuscript by Nadel et al. considers the effect of a single phosphorylation of a Tau fragment, TauC3, on its interaction with CHIP E3 protein ligase. They provide strong evidence that phosphorylation of S416 decreases the interaction and the ubiquitination of tauC3. This observation is of interest as the TauC3 fragment is known to accumulate in AD. The structure of the complex between CHIP and a peptide corresponding to TauC3 C-terminus is provided. Based on this structure, a CHIP mutation is designed and able to restore to some extent the binding of the phosphorylated fragment. The study is relevant as tau degradation is complex, including via the CHIP pathway that is mediated by the chaperone HSP70 but can also act without the need for the chaperone, depending of the tau isoform/fragment. Most importantly, the study reveals the complexity of phosphorylation as multiple tau phosphorylation can increase tau ubiquitination mediated by Hsp70.CHIP while a single phosphorylation of TauC3 decreases the ubiquitination mediated by CHIP alone. Identification of MARK2 as the kinase that phosphorylates S416 might be a simplification as they are probably other kinases able to phosphorylate this residue and that could be made clear. While the majority of the experiments have been performed in vitro, co-localization immuno-fluorescence experiments in human brain slices suggest that tauC3 is phosphorylated on S416 in AD patients. The manuscript is well written and results are presented in a logical manner. Methods are detailed.

>> Thank you for the kind words.

1. “we first confirmed that CHIP binds approximately 5-fold better to tauC3 over full length (FL) tau under basal conditions” : “binds 5-fold better” is not an appropriate characterization of the results as the dissociation constants have not been determined in this experiment – the signal is increased 5 times, the amount of complex is increased

>> We agree and have changed the wording to the more general “...tauC3 interacts with CHIP better than full length (FL) tau...” (pg 2).

2. Typo, supplementary figure 4 E : as derived from...missing information

>> Thanks for catching this error. We have fixed the legend to read: “...as derived from DSF experiments”.

3. tauC3p416 peptide would be of interest in experiments/graphs of Fig 2D-Fig S2A to facilitate comparison with the pseudo-phospho peptides

>> We agree and have repeated the DSF experiments using the full range of peptides, including the authentic phospho-peptide, tauC3 pS416 (new Fig 2D).

4. Fig 3A-B and Fig 1 E : a graphical representation for experiment 1E as in 3B would facilitate comparison of the 2 experiments/figures

>> We have added quantification of the *in vitro* ubiquitination blots in Fig S1D. As the reviewer suggested, this addition emphasizes how the rate of depletion of p.tauC3 and tauC3

S416E are roughly correlated (e.g., ~25 to 40% remaining after 5 minutes).

Reviewer 2.

In this study, the author reported that phosphorylation of tauC3 at a single residue, pS416, is sufficient to block its interaction with CHIP using many biochemical assay, crystal structure, mass spectrometry analysis, and imaging in clinical samples. This is an interesting and well-presented study. However, there were some defects and deficiencies in this study.

1. At first, the author hypothesized that HSP70-like mechanism might govern the binding of tauC3's C-terminal degron to CHIP. Why didn't the author compare HSP70 and Tau3C as control? In addition, the author should use the C-terminal peptide (phosphor-mimic 10mers) of HSP70 as control in Fig 2D-I. Finally, additional structural modeling of C-terminal peptide of HSP70 was necessary for comparison between HSP70 and Tau3C.

>> We agree that there is value in repeating the experiments that have documented the effects of pT636 phosphorylation on Hsp70 binding to CHIP (refs 37-38). In the revision, we have confirmed that CHIP binding is weakened when an Hsp70 10mer peptide is modified by either an authentic phosphorylation (pT636) or pseudo-phosphorylation (T636E), as measured by DSF (Fig 2C) and FP (Fig 2E, 2F). To further emphasize the origin of the hypothesis, we have added a visual sequence comparison of the two C-termini (Fig S2D) and a structural model of Hsp70's C-terminus bound to CHIP (PDB code 3Q49; Fig S2A-B). Together, we hope that these additions clarify why we originally hypothesized that tauC3 pS416 would interrupt CHIP binding. We thank the reviewer for the suggestions, as they help readers clearly understand this part of the narrative.

2. The author treated with the protein phosphatase inhibitor, okadaic acid (30 nM). However, there is no direct evidence that okadaic acid prevents dephosphorylation of Tau3C, especially S416. The references cited in this manuscript showed selective inhibition of PP2A by okadaic acid induced an AD-like hyperphosphorylation and accumulation of tau. Many of the sites that are dephosphorylated by PP2A are phosphorylated by either GSK-3 β or Cdk5. These include sites Ser199, Ser202, Thr205, Ser396, and Ser404. The author must show direct evidence of okadaic acid mediated dephosphorylation in Tau3C using mass spectrometry or western blot assay.

>> In the revision, we have highlighted the LC/MS data on the p.tauC3 purified from Sf9 cells after okadaic acid treatment (Fig 2SB), showing that 20 sites, including S416, are phosphorylated. In addition, we have added Western blots on the okadaic acid treated samples, showing multiple disease-associated phospho-forms of tauC3, including pS416 (Fig S2C). These data clarify that okadaic acid increases accumulation of pS416 on tauC3.

3. The author performed mass spectrometry analysis for the identification of phosphorylation and ubiquitinylation of tau isoforms. However, no mass spec data provided. The author should provide the detailed information of identification of phosphorylation and ubiquitinylation including database search score, FDR level, localization score for PTM sites, etc. In addition, mass spectrometry raw data should be deposited in public repository.

>> We apologize for the oversight. The raw mass spectrometry data is now deposited in PRIDE (accession code: PXD052540). Additionally, the peptide data is summarized in Table S1 and additional details on FDR and SLIP score criteria is included in the revised methods.

Reviewers can access the mass spectrometry data: <https://www.ebi.ac.uk/pride/review-dataset/8f9401e858354d818c1631cea577c203>.

Alternatively, they can use:

Username: reviewer_pxd052540@ebi.ac.uk

Password: sV4ExAJH7CAv

4. In Fig 5, the author observed that significant increases in both tau pS416 and tauC3 across disease progression. However, high correlation between pS416 and tauC3 also suggests the possibility that phosphorylation may simply increase due to an increase in the amount of Tau3C protein. Did the author check the amount of FL tau using immunofluorescence?

>> The high correlation between the accumulation of pS416 and tauC3 certainly suggests, as the reviewer states, that tauC3 pS416 is accumulating over time as the level of protein increases. Indeed, we reached this same conclusion. The other possibility, that the reviewer also briefly suggests, is that the pS416 signal is derived from this PTM being present on other tau proteoforms, such as FL tau. Unfortunately, there is not a reliable IF antibody for human FL tau, so we were not able to conduct this comparison directly. Accordingly, we have further emphasized the caveats with the experiment in the revised Discussion.

5. The author suggested that MARK2-mediated phosphorylation of tauC3 at pS416 is sufficient to block binding to CHIP using in vitro and cellular assays. However, the author performed only in-vitro assay using recombinant MARK2 and tau isoforms. The reference paper cited by author showed that multiple sites are phosphorylated as well as S416. It is difficult to confirm a direct relationship between MARK2 and S416 based on these results alone. At a minimum, results from in vivo assay (MARK2 knockout, etc.) are required. In addition, if the author was going to do an in vitro assay, why didn't the author experiment with CAMKII?

>> In the revision, we have further clarified the fact that both MARK2 and CAMKII are already well-known to be responsible for phosphorylation at pS416 (amongst other sites; see revised text on pages 5 and 6). We did not intend to imply any special role for MARK2, but focused on this kinase because, as outlined in the text, CAMKII is not expressed in our particular cell models. Rather, the point of this work is to identify, using structural and biochemical studies, that pS416 is the most important phospho-site for CHIP binding to tauC3. Of course, this conclusion does not rule out roles for other MARK2 or CAMKII sites elsewhere in tau, especially in a different context or in binding to other partners. We thank the reviewer for the opportunity to clarify these points.

6. The author reported that CHIP binds p.tau ~10-fold better than it binds to unmodified tau. These findings show that TPR domain of CHIP recognizes a phosphorylation-dependent degron on tau centered on pS214 and pT212. The author should discuss differences between this study and the author's recent report.

>> We appreciate this comment. Indeed, one of our major goals was to make it clear that CHIP is capable of binding a wide range of tau proteoforms (e.g., FL p-tau, tauC3, etc.) using different binding sites/modes. The fascinating observation, as nicely summarized by the reviewer, is that some of those sites bind better when they are phosphorylated (e.g., the internal phospho-degron on FL p-tau), while others are significantly weakened by phosphorylation (e.g., CHIP binding to pS416 tauC3). Perhaps this idea is best shown in Figure 1B. More broadly, this

diversity of mechanisms is likely why CHIP is such a strong regulator of tau proteostasis, a hypothesis first articulated by Len Petrucelli, Chad Dickey and others. We have made changes throughout the Discussion to better highlight this concept and hope that these alterations clarify the broader context of the current work.

Reviewer 3.

In this manuscript Nadel et al provide a beautiful characterization of the effect of phosphorylation of tau3C isoform on its interaction with CHIP E3 ligase, and its subsequent cellular ubiquitination and degradation.

Through an elegant combination of structural biology, biophysics and cellular assays, the authors show that a single phosphorylation in tau3C is sufficient to inhibit the CHIP-tauC3 interaction, thus providing a mechanistic explanation as to why these species accumulate in the brains of AD patients.

The manuscript is technically excellent and was a pleasure to read. I have but a few minor comments –

>> Thank you for the supportive comments.

1) The authors state on page 3 that phosphorylation masks CHIP ubiquitination sites on tau3C, however supplementary figure 1c does not show any sites that were found to be ubiquitinated only in tau3C and not p.tauC3. If such sites were indeed identified, they should be clearly shown in that figure, otherwise the statement needs to be rephrased.

>> We did not mean to imply that phosphorylation directly impacts ubiquitination and agree that the previous wording could be misinterpreted. We have replaced that sentence with a much more general conclusion: that CHIP is able to ubiquitinate both tauC3 and p.tauC3. Finally, to answer the reviewer's specific question, there are no sites identified on tauC3 that were not present on p.tauC3.

2) While the evidence that phosphorylation of tauC3 weakens binding to CHIP is very clearly shown, the hindered ubiquitination ability is less so. While the ubiquitination is reduced, it still is present at a significantly higher degree for p.tau3C or tau3C S416E mutant compared to the regular tau. Is this just a matter of CHIP's affinity for tau? If this is the case, no ubiquitination should be observed for tau3C lacking the 6-7 most C-terminal residues.

>> This is a good point. The effect of pS416 on tauC3's affinity for CHIP is between ~5-fold (Fig 2E) and 10-fold (Fig 2J), leading to a reduce, but not ablated ubiquitination rate. In contrast, mutation of the terminal aspartic acid in tauC3 to alanine leads to a complete loss in affinity for CHIP ($K_d > 50 \mu\text{M}$) and, as predicted by the reviewer, nearly complete loss of ubiquitination (ref 36). Thus, we entirely agree that the degree of *in vitro* ubiquitination seems to roughly correlate with relative, measured affinity. At the same time, the accumulation of pS416 tauC3 in cells (see Fig 3C) patient brains (see Fig 5A) seems to suggest that even a modest loss of affinity can, over time, lead to meaningful changes in CHIP-mediated degradation. This interplay between affinity and ubiquitination is perhaps not surprising when one considers the complex mechanisms of ubiquitin transfer. We have elaborated on this important idea in the revised text (see page 4).

3) The authors imply that the C-terminal region of tau3C is its sole binding site to CHIP. However, phosphorylation of this region only slightly reduces the affinity, and p.tau3C still binds CHIP significantly stronger than the regular tau isoform. Can the authors rule out the presence of other binding sites on CHIP for tau3C?

>> As the reviewer says, one can never rule out the contribution of secondary binding sites and we did not mean to imply that they aren't important. To explore this idea in more detail we have performed additional ELISA studies to measure CHIP binding to FL tau and FL p.tau (Fig 1C). The results confirm that tauC3 is the tightest binding proteoform (Kd ~190 nM) and, moreover, that CHIP's affinity for either FL p.tau (Kd ~500 nM) or p.tauC3 (Kd ~700 nM) is weakened by 2- to 3-fold. Because these differences in affinity are significant, but modest, it is likely that secondary binding sites (e.g, not the C-terminus) also contribute to CHIP's interaction with p.tauC3. Indeed, we would argue that this result is entirely expected, based on our previous work (ref 23 and 36). We have added text to discuss this point and make it clearer that secondary sites contribute additional binding energy (pg. 3).

4) The effect of the phosphorylation on the cellular degradation of tau3C is very minor. Would this effect be better seen by CHX chase assays, for example?

>> We have modified the wording to better reflect that the impact of tauC3 S416E is statistically significant, but relatively modest (Fig 3G). Indeed, we did not originally expect this difference to be dramatic because the S416E modification does not fully prevent processive, CHIP-mediated ubiquitination *in vitro* (see Fig 3A and response above). Moreover, as noted in the manuscript, there appears to be a feedback loop in these cells, because the tauC3 mRNA is slightly elevated for the S416E variant (Fig 3H). Yet, it is possible that, over long periods of time, even a 30% change in tauC3 stability/levels could create a substantial impact. Unfortunately, the suggested CHX chase experiments do not work well for tau because the lifetime of the protein (measured in days to weeks) is significantly longer than the division time of the cells. We thank the reviewer for helping us clarify these expectations.

Reviewer #4

The quality of the crystallographic structures presented in this manuscript is good overall. It is possible that the data have been processed a little too conservatively - a CC(1/2) value of 0.57 in the highest resolution shell is very high - but processing pipelines such as xia2 tend towards the conservative by default and this is not a serious flaw with the data. The quality of the models presented is good and the figures are clear.

>> We thank the reviewer for the supportive comments. In internal discussions, we indeed opted to use a relatively conservative data processing approach and appreciate that the reviewer is understanding.

Reviewer #5

This manuscript by Nadel et al. describes the interaction between phosphorylation and ubiquitination of cleaved tau as it relates to the E3 ubiquitin ligase, CHIP. The authors provide biochemical, biophysical, and structural data to support their conclusions. Overall, the data in this manuscript are rigorous and provide new mechanistic insight into the competition of tau posttranslational modifications that regulate its accumulation or degradation, respectively. However, there are some claims made that should be softened and some additional experiments and discussion that would further strengthen the manuscript.

1. It is unclear how much of an impact this competition over phosphorylation/ubiquitination has on tau. How does this competition affect tau aggregation, degradation, or other outcomes?

>> As mentioned above, we have removed the sentence that inadvertently implied a possible interplay between phosphorylation and ubiquitination, which was not the intent.

2. Does ubiquitination by CHIP inversely block the phosphorylation of tau at S416?

>> This is an interesting idea, because it might create a feed-forward loop in which CHIP-mediated ubiquitination would help the E3 ligase to partially circumvent the effects of pS416 phosphorylation. We hope to explore this idea in future work, after first identifying the relevant E2 ligases so that the physiological ubiquitination patterns can be used.

3. While the S416E did significantly affect CHIP binding, the binding was fully blocked. The authors should at least discuss what other factors may be contributing to the interaction outside of this one phosphorylation site.

>> We appreciate this comment, which aligns with those of Reviewers 2 and 3. Briefly, as noted above, both our data and published reports (ref 36) suggest that, as this Reviewer implies, relative affinity for CHIP partially predicts the extent of ubiquitination *in vitro*. As mentioned in response to the comments above, we have clarified these points in the revision.

4. Recombinant tau is challenging to isolate without having some Hsp70 still associated. What is the purity of tau? Is there Hsp70 present from the protein production? Does this Hsp70 have an impact on the effects of CHIP on tau?

>> We have added a Coomassie gel to show that the p.tau protein used here is >90% pure with no obvious band for Hsp70 isoforms (Fig S1A). Perhaps more importantly, one would expect that Hsp70 contamination would impact the measured affinity for CHIP; whereas, the observed affinity of CHIP for purified tauC3 (Kd ~190 nM) matches well with its affinity for the wholly synthetic, tauC3 C-terminal peptides (Kd ~150 nM). Together, these quality assurance steps convinced us that any Hsp70 contamination must be minor / insignificant.

5. In a prior manuscript by this group in 2018 in NSMB, it was demonstrated that CHIP interacts most with 3R tau. Are the effects on 3R0N tau in line with the 4R0N tau?

>> First, to clarify, the 2018 data shows that CHIP is better at suppressing aggregation of 0N3R tau compared to 0N4R tau (and does not explore binding affinity *per se*). Here, because the measured affinity of CHIP for tauC3 protein matches well with the measured affinity for the C-terminal 10mer peptide (as discussed above), we conclude that binding site(s) outside the C-terminus are a relatively minor contributor and, thus, that any contribution of 3R vs. 4R will be minor. That being said, it will be interesting to specifically study the role of CHIP in 3R vs 4R tau proteostasis in future studies.

6. The data from human brain tissue is on a limited set of patients. It may be more convincing to include CHIP and ubiquitin co-staining.

>> In response to a similar comment from Reviewer 2, we have added data from an additional set of patients (Fig S5), which further supports the conclusions. We hope that future

efforts will expand on these studies to explore the relative levels of CHIP, ubiquitination and various tau proteoforms.

7. Does the ubiquitination of tau that reacts with S416 antibody change?

>> We are not entirely clear what specific experiment is being suggested. That being said, we do not see evidence for an increase in ubiquitination of pS416 tauC3 in the cellular models (see Fig 3E). Moreover, in the *in vitro* ubiquitination experiments, tauC3 S416E was a relatively poor substrate for CHIP (see Fig 3A).

8. All western blots should include quantification.

>> We have added quantifications of Western blots (see Fig S6).

9. In addition to regulating tau turnover, it has been shown that ubiquitination can help promote the fibrilization of tau. The authors should discuss this in the context of their findings.

>> In our experience and in the literature (and seemingly in the reviewer's experience too), ubiquitination of tau can have a wide variety of effects, including promoting or inhibiting aggregation and driving formation of amorphous aggregates. *In vivo*, it seems likely that the combination of E2s, E4s and DUBs will guide tau ubiquitination outcomes in ways that are not necessarily captured by cellular or *in vitro* models. We have added a sentence to make this point clearer in the Discussion.

10. Limitations of the work should be included.

>> In addition to softening the language around the conclusions (as mentioned above), we added a specific Discussion statement about the need for validation in relevant models of disease.

11. Some bold claims from the data, which shows partial blocks/reductions should be softened to better represent the data.

>> We agree, especially concerning the effects of pS416 on the affinity for CHIP, which we originally overstated. We have systematically edited the text, with the goal of replacing words such as "inhibit" and "block" with "weaken" or "limit". We thank the reviewer for the comment.

12. There appears to be an error in Figure 6 figure legend in reference to the quant of panel C.

>> Thank you! The error has been corrected.

13. All raw, uncut blots should be shown.

>> Raw blots have been added (Fig S6).

Reviewers' Comments:

Reviewer #1:

Remarks to the Author:

The authors have provided a thoroughly revised version of their manuscript, including the addition of raw data, new experiments and some nuances in their original conclusions regarding a very complex mechanism, related to tau multiple phosphorylation, that they have been able to decipher. The manuscript is well written and I am confident that the conclusions are based on sound experimental work.

Reviewer #2:

Remarks to the Author:

The authors adequately revised the manuscript according to the suggestion of the reviewer.

As minor comment, in page 3 line 118 and line 141, I think it is Supplementary data S1, but it is written as Supplementary 1B and Supplementary S1B, respectively.

Reviewer #3:

Remarks to the Author:

The authors have addressed all my comments. I have no further concerns

Reviewer #5:

Remarks to the Author:

The reviewers have addressed all of my prior concerns through added/edited text and some additional figures. They should be applauded for their efforts in addressing each of the concerns. I have no additional comments.